# Sparse Meets Dense: Unified Generative Recommendations with Cascaded Sparse-Dense Representations

**Yuhao Yang, Zhi Ji,** * **Zhaopeng Li, Yi Li, Zhonglin Mo,**
**Yue Ding, Kai Chen, Zijian Zhang, Jie Li, Shuanglong Li, Lin Liu**
Baidu Inc.,
Beijing, China
{yangyuhao01, jizhi}@baidu.com

## Abstract

Generative models have recently gained attention in recommendation systems by directly predicting item identifiers from user interaction sequences. However, existing methods suffer from significant information loss due to the separation of stages such as quantization and sequence modeling, hindering their ability to achieve the modeling precision and accuracy of sequential dense retrieval techniques. Integrating generative and dense retrieval methods remains a critical challenge. To address this, we introduce the Cascaded Organized Bi-Represented generAtive retrieval (COBRA) framework, which innovatively integrates sparse semantic IDs and dense vectors through a cascading process. Our method alternates between generating these representations by first generating sparse IDs, which serve as conditions to aid in the generation of dense vectors. End-to-end training enables dynamic refinement of dense representations, capturing both semantic insights and collaborative signals from user-item interactions. During inference, COBRA employs a coarse-to-fine strategy, starting with sparse ID generation and refining them into dense vectors via the generative model. We further propose BeamFusion, an innovative approach combining beam search with nearest neighbor scores to enhance inference flexibility and recommendation diversity. Extensive experiments on public datasets and offline tests validate our method's robustness. Online A/B tests on a real-world advertising platform with over 200 million daily users demonstrate substantial improvements in key metrics, highlighting COBRA's practical advantages.

## 1 Introduction

Recommendation systems are vital components of modern digital ecosystems, providing personalized item suggestions that align with user preferences across e-commerce platforms, streaming services, and social networks [1–3]. Recent advancements have focused on sequential recommendation methods, which leverage the sequential nature of user interactions to enhance recommendation performance [4–7]. Notable models like SASRec [8] and BERT4Rec [9] have demonstrated the effectiveness of sequence models in capturing user behavior patterns.

The emergence of generative models has further expanded the capabilities of recommendation systems [10–12]. Unlike traditional sequential recommendation methods, generative models can directly predict target items based on user behavior sequences [13–15]. These models handle complex user-item interactions and offer emerging abilities such as reasoning and few-shot learning, which significantly improve recommendation accuracy and diversity [16–18]. Among these, TIGER [19]

---

* Corresponding author.

39th Conference on Neural Information Processing Systems (NeurIPS 2025).

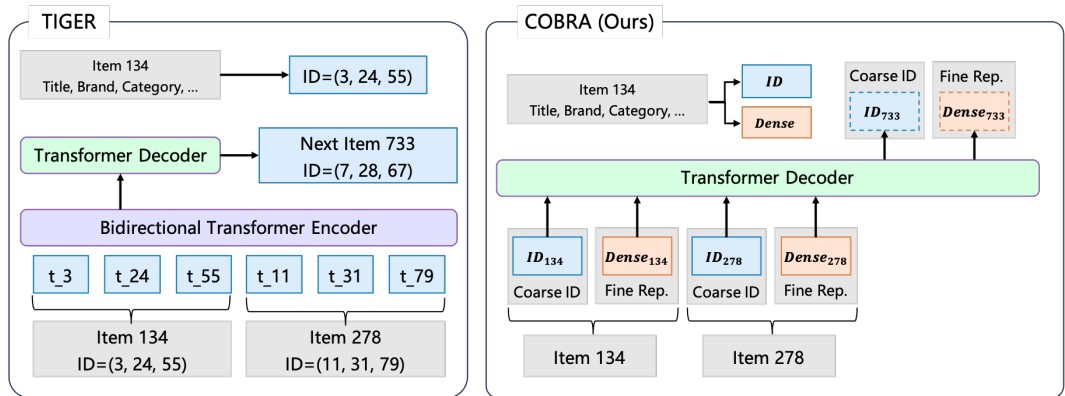

Figure 1: Comparison of generative recommendation paradigms. The left side illustrates traditional generative retrieval approaches, exemplified by TIGER, which uses a sequence of sparse IDs as input within a Transformer encoder-decoder architecture to directly predict the sparse ID of the next item. The right side depicts the proposed COBRA(Cascaded Organized Bi-Represented generAtive retrieval), which integrates sparse IDs for coarse semantics and dense vectors for fine details. The cascaded representation is processed by a Transformer decoder that sequentially predicts the sparse ID followed by the dense vector.

is a pioneering approach in generative retrieval for recommendation systems. As depicted in Figure 1(Lower Left), TIGER leverages a Residual Quantized Variational AutoEncoder (RQ-VAE) [20] to encode item content features into hierarchical semantic IDs, allowing the model to share knowledge across semantically similar items without the need for individual item embeddings. Beyond TIGER, several other methods have been proposed to further explore the integration of generative models with recommendation systems. LC-Rec [21] aligns semantic and collaborative information using RQ-VAE with a series of alignment tasks. ColaRec [22] combines collaborative filtering signals with content information by deriving generative identifiers from a pretrained recommendation model. IDGenRec [23] leverages large language models to generate unique, concise, and semantically rich textual identifiers for recommended items, showing strong potential in zero-shot settings.

Despite these innovations, existing generative recommendation methods still face several challenges compared to sequential dense retrieval methods [24, 25]. Sequential dense retrieval methods, which rely on dense embeddings for each item, offer high accuracy and robustness but require substantial storage and computational resources. In contrast, generative methods, while efficient, often struggle with fine-grained similarity modeling [26]. To effectively leverage the strengths of both retrieval paradigms, we propose Cascaded Organized Bi-Represented generAtive retrieval(COBRA), a framework that synergizes generative and dense retrieval. Figure 1(Right) illustrates the cascaded sparse-dense representations in COBRA. The proposed method introduces a cascaded generative retrieval framework alternating between generating sparse IDs and dense vectors. This approach mitigates the information loss inherent in ID-based methods. Specifically, COBRA's input is a sequence of cascaded representations composed of sparse IDs and dense vectors corresponding to items in the user's interaction history. During training, dense representations are learned through contrastive learning objectives in an end-to-end manner. By first generating the sparse ID and then the dense representation, COBRA reduces the learning difficulty of dense representations and promotes mutual learning between the two representations. During inference, COBRA employs a coarse-to-fine generation process, starting with sparse ID that provides a high-level categorical sketch capturing the categorical essence of the item. The generated ID is then appended to the input sequence and fed back into the model to predict the dense vector that captures the fine-grained details, enabling more precise and personalized recommendations. To ensure flexible inference, we introduce BeamFusion, a sampling technique combining beam search with nearest neighbor retrieval scores, ensuring controllable diversity in the retrieved items. Unlike TIGER, which relies solely on sparse IDs, COBRA harnesses the strengths of both sparse and dense representations.

Our main contributions are as follows:

- **Cascaded Bi-Represented Retrieval Framework**: We introduce COBRA, a framework that alternates between generating sparse semantic IDs and dense vectors. It addresses information loss in ID-based methods and reduces the difficulty of representation learning by using sparse IDs as conditions for generating dense vectors.

- **Learnable Dense Representations via End-to-End Training**: COBRA uses the original item data as input to generate dense representations through end-to-end training. Unlike static embeddings, COBRA's dense vectors are dynamically learned, capturing semantic information and fine-grained details.

- **Coarse-to-Fine Generation Process**: During inference, COBRA first generates sparse IDs, which are then fed back into the model to produce refined dense representations, enhancing the granularity of the dense vectors. We also introduce BeamFusion for more diverse recommendations.

- **Comprehensive Empirical Validation**: Extensive experiments on benchmark datasets show COBRA surpasses current methods in recommendation accuracy, proving its effectiveness in balancing precision and diversity.

## 2 Related Work

**Sequential Dense Recommendation.** Early sequential recommendation systems leveraged RNNs and CNNs to model user behavior sequences [27, 28]. The introduction of Transformer-based methods, such as SASRec [8] and BERT4Rec [9], greatly improved the capability to capture complex user dynamics. Recent models focus on cross-domain transferability and the integration of textual features through contrastive learning [29–31].

**Generative Recommendation.** Generative approaches have shifted the field from discriminative ranking to directly generating item identifiers [32, 19, 33]. Some models treat recommendation as a language modeling task [33], while others generate semantically meaningful or structured identifiers [19, 21, 34]. Hybrid methods, such as LIGER [26], combine generative and dense retrieval to overcome the limitations of each approach. However, how to more flexibly integrate these paradigms remains an open problem. A more comprehensive review is provided in Appendix A.

## 3 Methodology

This section introduces the Cascaded Organized Bi-Represented generAtive Retrieval (COBRA) framework, which integrates cascaded sparse-dense representations and coarse-to-fine generation to enhance recommendation performance. Figure 2 illustrates the overall framework of COBRA.

### 3.1 Sparse-Dense Representation

**Sparse Representation.** COBRA generates sparse IDs using a Residual Quantized Variational Autoencoder (RQ-VAE), inspired by the approach in TIGER [19]. For each item, we extract its attributes to generate a textual description, which is embedded into a dense vector space and quantized to produce sparse IDs. These IDs capture the categorical essence of items, forming the basis for subsequent processing. For the sake of brevity, the subsequent methodology descriptions will assume that the sparse ID consists of a single level. However, it should be noted that this approach can be easily extended to accommodate scenarios involving multiple levels.

**Dense Representation.** To capture nuanced attribute information, we develop an end-to-end trainable dense encoder, encoding item textual contents. Each item's attributes are flattened into a text sentence, prefixed with a [CLS] token, and fed into a Transformer-based text encoder **Encoder**. The dense representation $\mathbf{v}_t$ is extracted from the output corresponding to the [CLS] token, capturing fine-grained details of the item's textual content. As illustrated in the lower part of Figure 2, we incorporate position embeddings and type embeddings to model the positional and context of tokens within the sequence. These embeddings are added to the token embeddings, enhancing the model's ability to distinguish between different tokens and their positions in the sequence.

**Cascaded Representation.** The cascaded representation integrates sparse IDs and dense vectors within a unified generative model. Specifically, for each item, we combine its sparse ID $ID_t$ and dense vector $\mathbf{v}_t$ to form a cascaded representation $(ID_t, \mathbf{v}_t)$. This approach leverages the strengths of both representations, providing a more comprehensive characterization of items: sparse IDs provide

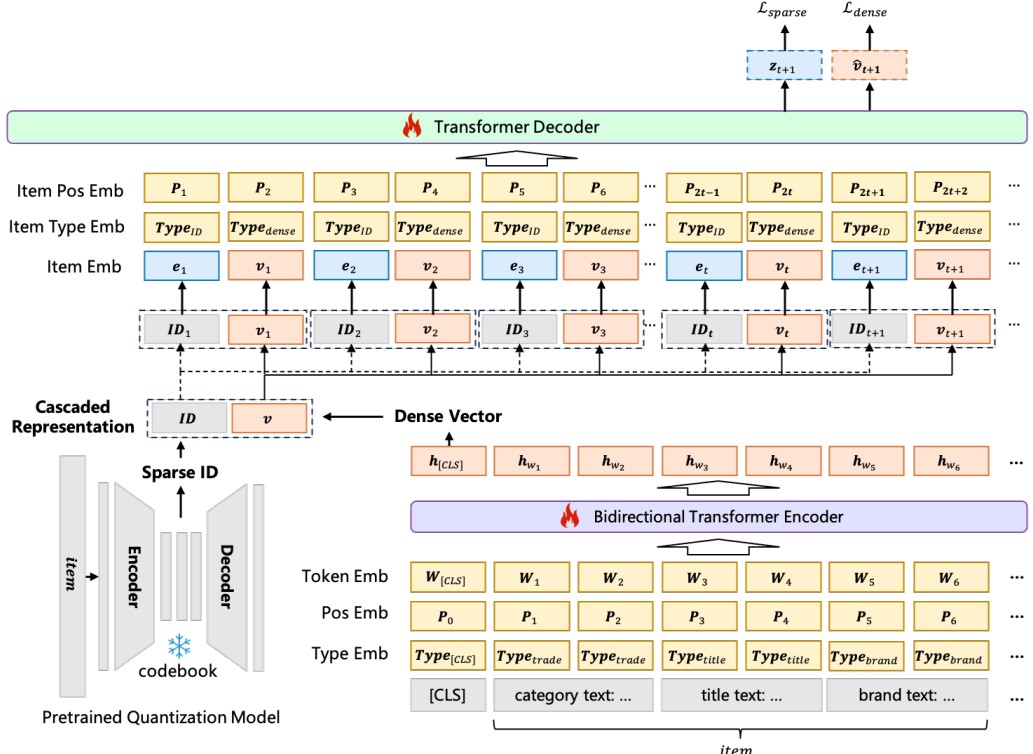

Figure 2: The architecture of COBRA. The model employs a cascaded sparse-dense representation approach, where sparse IDs are generated via Residual Quantization and dense vectors are produced by a trainable Transformer Encoder. These representations serve as inputs to a Transformer Decoder, which alternates between predicting sparse IDs and dense vectors. The predicted outputs are used to compute the loss functions $\mathcal{L}_{\text{sparse}}$ and $\mathcal{L}_{\text{dense}}$. For the sake of simplicity, the figure illustrates an example with a single level of sparse ID.

a stable categorical foundation through discrete constraints, while dense vectors maintain continuous feature resolution, ensuring that the model captures both high-level semantics and fine-grained details.

## 3.2 Sequential Modeling

**Probabilistic Decomposition.** The probability distribution modeling of the target item is factorized into two stages, leveraging the complementary strengths of sparse and dense representations. Specifically, instead of directly predicting the next item $s_{t+1}$ based on the historical interaction sequence $S_{1:t}$, COBRA predicts the sparse ID $ID_{t+1}$ and the dense vector $\mathbf{v}_{T+1}$ separately:

$$P(ID_{t+1}, \mathbf{v}_{t+1}|S_{1:t}) = P(ID_{t+1}|S_{1:t})P(\mathbf{v}_{t+1}|ID_{t+1}, S_{1:t}) \tag{1}$$

where $P(ID_{t+1}|S_{1:t})$ represents the probability of generating the sparse ID $ID_{t+1}$ based on the historical sequence $S_{1:t}$, capturing the categorical essence of the next item. $P(\mathbf{v}_{t+1}|ID_{t+1}, S_{1:t})$ represents the probability of generating the dense vector $\mathbf{v}_{t+1}$ given the sparse ID $ID_{t+1}$ and the historical sequence $S_{1:t}$, capturing the fine-grained details of the next item. This decomposition allows COBRA to leverage both the categorical information provided by sparse IDs and the fine-grained details captured by dense vectors.

**Sequential Modeling with a Unified Generative Model.** For sequential modeling, we utilize a unified generative model based on the Transformer architecture to effectively capture sequential dependencies in user-item interactions. The Transformer receives an input sequence of cascaded representations, with each item represented by its sparse ID and dense vector.

The sparse ID, denoted as $ID_t$, is transformed into a dense vector space through an embedding layer: $\mathbf{e}_t = \text{Embed}(ID_t)$. This embedding $\mathbf{e}_t$ is concatenated with the dense vector $\mathbf{v}_t$ to form the model's input at each time step: $\mathbf{h}_t = [\mathbf{e}_t; \mathbf{v}_t]$.

Our Transformer Decoder model comprises multiple layers, each featuring self-attention mechanisms and feedforward networks. As depicted in the upper part of Figure 2, the input sequence to the Decoder consists of cascaded representations. To enhance modeling of sequential and contextual information, these representations are augmented with item position and type embeddings. For brevity, mathematical formulations in the following sections focus on the cascaded sequence representation, omitting explicit notation for position and type embeddings. The Decoder processes this enriched input to generate contextualized representations for predicting the subsequent sparse ID and dense vector.

**Sparse ID Prediction.** Given history interaction sequence $S_{1:t}$, to predict the sparse ID $ID_{t+1}$, the Transformer input sequence is: $\mathbf{S}_{1:t} = [\mathbf{h}_1, \mathbf{h}_2, \ldots, \mathbf{h}_t] = [\mathbf{e}_1, \mathbf{v}_1, \mathbf{e}_2, \mathbf{v}_2, \ldots, \mathbf{e}_t, \mathbf{v}_t]$. where each $\mathbf{h}_i$ is a concatenation of the sparse ID embedding and the dense vector for the $i$-th item. The Transformer processes this sequence to generate contextualized representations, subsequently used to predict the next sparse ID and dense vector. Specifically, the Transformer decoder processes the sequence $\mathbf{S}_{1:t}$, producing a sequence of vectors $\mathbf{y}_t = \text{TransformerDecoder}(\mathbf{S}_{1:t})$. The logits for sparse ID prediction are derived as: $\mathbf{z}_{t+1} = \text{SparseHead}(\mathbf{y}_t)$. where $\mathbf{z}_{t+1}$ represents the logits for the predicted sparse ID $ID_{t+1}$.

**Dense Vector Prediction.** For predicting the dense vector $\mathbf{v}_{t+1}$, the Transformer input sequence can be represented as: $\bar{\mathbf{S}}_{1:t} = [\mathbf{S}_{1:t}, \mathbf{e}_{t+1}] = [\mathbf{e}_1, \mathbf{v}_1, \mathbf{e}_2, \mathbf{v}_2, \ldots, \mathbf{e}_t, \mathbf{v}_t, \mathbf{e}_{t+1}]$. The Transformer decoder processes $\bar{\mathbf{S}}_{1:t}$ to output the predicted dense vector: $\hat{\mathbf{v}}_{t+1} = \text{TransformerDecoder}(\bar{\mathbf{S}}_{1:t})$.

### 3.3 End-to-End Training

In COBRA, the end-to-end training process is designed to optimize both sparse and dense representation prediction jointly. The training process is governed by a composite loss function that combines losses for sparse ID prediction and dense vector prediction.

**Sparse ID Loss.** The sparse ID prediction loss, denoted as $\mathcal{L}_{\text{sparse}}$, ensures the model's proficiency in predicting the next sparse ID based on the historical sequence $S_{1:t}$:

$$\mathcal{L}_{\text{sparse}} = -\sum_{t=1}^{T-1} \log \left( \frac{\exp(z_{t+1}^{ID_{t+1}})}{\sum_{j=1}^{C} \exp(z_{t+1}^{j})} \right) \tag{2}$$

where $T$ is the length of the historical sequence, $ID_{t+1}$ is the sparse ID corresponding to interacted item at time step $t+1$, $z_{t+1}^{ID_{t+1}}$ represents the predicted logit of ground truth sparse ID $ID_{t+1}$ at time step $t+1$, generated by the Transformer Decoder, and $C$ denotes the set of all sparse IDs.

**Dense vector Loss.** The dense vector prediction loss $\mathcal{L}_{\text{dense}}$ focuses on refining the dense vectors, enabling them to discern between similar and dissimilar items. The loss is defined as:

$$\mathcal{L}_{\text{dense}} = -\sum_{t=1}^{T-1} \log \frac{\exp(\cos(\hat{\mathbf{v}}_{t+1}, \mathbf{v}_{t+1}))}{\sum_{item_j \in \text{Batch}} \exp(\cos(\hat{\mathbf{v}}_{t+1}, \mathbf{v}_{item_j}))} \tag{3}$$

where $\hat{\mathbf{v}}_t$ is the predicted dense vector, $\mathbf{v}_t$ is the ground truth dense vector for the positive item, and $\mathbf{v}_{item_j}$ represents the dense vectors of items within the batch. The term $\cos(\hat{\mathbf{v}}_{t+1} \cdot \mathbf{v}_{t+1})$ represents the cosine similarity between the predicted and ground truth dense vectors. The dense vectors are generated by an end-to-end trainable encoder denoted by **Encoder**, which is optimized during the training process. This ensures that the dense vectors are dynamically refined and adapted to the specific requirements of the recommendation task.

**Overall Loss.** The overall loss function is formulated as: $\mathcal{L} = \mathcal{L}_{\text{sparse}} + \mathcal{L}_{\text{dense}}$. The dual-objective loss function enables a balanced optimization process, where the model dynamically refines dense vectors guided by sparse IDs. This end-to-end training approach captures both high-level semantics and feature-level information, optimizing sparse and dense representations jointly for improved performance.

### 3.4 Coarse-to-Fine Generation

During the inference phase, COBRA implements the coarse-to-fine generation procedure, involving the sequential generation of sparse IDs followed by the refinement of dense vectors in a cascaded manner, as illustrated in Figure 3. The coarse-to-fine generation process in COBRA is designed to capture both the categorical essence and fine-grained details of user-item interactions. This process involves three main stages:

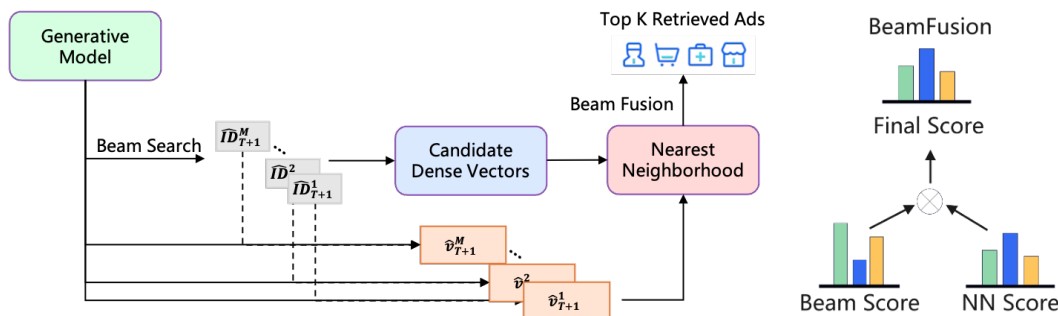

Figure 3: Illustration of the Coarse-to-Fine Generation process. During inference, $M$ sparse IDs are generated via Beam Search, and appended to the sequence. Dense vectors are then generated and used in ANN to obtain candidate items. BeamFusion combines beam scores and similarity scores to rank candidates, from which the top $K$ items are selected.

**Sparse ID Generation.** Given a user sequence $S_{1:T}$, we utilize the ID probability distribution modeled by the Transformer Decoder, $\hat{ID}_{T+1} \sim P(i_{T+1}|S_{1:T})$, and employ the BeamSearch algorithm to derive the top $M$ IDs. The formulation is as follows:

$$\{\hat{\mathbf{ID}}_{T+1}^{k}\}_{k=1}^{M} = \text{BeamSearch}(\text{TransformerDecoder}(\mathbf{S}_{1:T}), M) \tag{4}$$

where $k \in \{1, 2, \ldots, M\}$. Each generated ID is associated with a beam score $\phi_{\hat{\mathbf{ID}}_{T+1}^{k}}$.

**Dense Vector Refinement.** Each generated sparse ID $\hat{\mathbf{ID}}_{T+1}^{k}$ is subsequently converted into an embedding and appended to the previous cascaded sequence embedding $\mathbf{S}_{1:T}$. Then the corresponding dense vector $\hat{\mathbf{v}}_{T+1}^{k}$ is generated:

$$\hat{\mathbf{v}}_{T+1}^{k} = \text{TransformerDecoder}([\mathbf{S}_{1:T}, \text{Embed}(\hat{\mathbf{ID}}_{T+1}^{k})]) \tag{5}$$

After that, we employ Approximate Nearest Neighbor (ANN) search to retrieve the top $N$ candidate items:

$$\mathcal{A}_{k} = \text{ANN}(\hat{\mathbf{ID}}_{T+1}^{k}, \mathcal{C}(\hat{\mathbf{ID}}_{T+1}^{k}), N) \tag{6}$$

where $\mathcal{C}(\hat{\mathbf{ID}}_{T+1}^{k})$ is the set of candidate items associated with sparse ID $\hat{\mathbf{ID}}_{T+1}^{k}$, and $N$ represents the number of top items to be retrieved.

**BeamFusion Mechanism.** In order to achieve a balance between precision and diversity, we devise a globally comparable score for items corresponding to each sparse ID. This score is capable of reflecting both the differences among different sparse IDs and the fine-grained difference among items under the same sparse ID. To accomplish this, we propose the BeamFusion mechanism:

$$\Phi^{(\hat{\mathbf{v}}_{T+1}^{k}, \hat{\mathbf{ID}}_{T+1}^{k}, \mathbf{a})} = \text{Softmax}(\tau\phi_{\hat{\mathbf{ID}}_{T+1}^{k}}) \times \text{Softmax}(\psi\cos(\hat{\mathbf{v}}_{T+1}^{k}, \mathbf{a})) \tag{7}$$

where $\mathbf{a}$ represents the candidate item, $\tau$ and $\psi$ are coefficients, and $\phi_{\hat{\mathbf{ID}}_{T+1}^{k}}$ denotes the beam score obtained during the beam search process. Finally, we rank all candidate items based on their BeamFusion Scores and select the top $K$ items as the final recommendations:

$$\mathcal{R} = \text{TopK}\left(\bigcup_{k=1}^{M} \mathcal{A}_{k}, \Phi, K\right) \tag{8}$$

where $\mathcal{R}$ denotes the set of final recommendations, and $\text{TopK}(\cdot)$ represents the operation of selecting the top $K$ items with the highest BeamFusion Scores. For a detailed algorithmic description, please refer to the pseudocode provided in Appendix E.

### 3.5 Theoretical Justification

In our framework, each item is characterized by a hybrid sparse-dense representation $(ID, \mathbf{v})$, where $ID$ denotes the sparse ID and $\mathbf{v}$ represents the dense vector. A critical consideration is how to model the joint conditional distribution $P(ID, \mathbf{v}|S)$ given a user sequence $S$. This distribution can be modeled in two primary ways:

**Independent Modeling**  This approach assumes that $ID$ and $\mathbf{v}$ are predicted independently given $S$:

$$P(ID, \mathbf{v}|S) = P(ID|S) \cdot P(\mathbf{v}|S). \tag{9}$$

**Cascaded Modeling (Ours)**  This approach, which we adopt, factorizes the joint distribution by first predicting $ID$, and subsequently predicting $\mathbf{v}$ conditioned on both $ID$ and $S$:

$$P(ID, \mathbf{v}|S) = P(ID|S) \cdot P(\mathbf{v}|ID, S). \tag{10}$$

We posit that the cascaded formulation is theoretically superior as it explicitly captures the dependency between $ID$ and $\mathbf{v}$ given $S$. We formalize this advantage in terms of information entropy.

**Theorem 1** (Superiority of Cascaded Modeling). *Let $H_{indep}$ and $H_{cascaded}$ denote the total entropies of the probability distributions defined by the independent (Eq. 9) and cascaded (Eq. 10) formulations, respectively. Then,*

$$H_{cascaded}(ID, \mathbf{v}|S) \leq H_{indep}(ID, \mathbf{v}|S) \tag{11}$$

*with equality holding if and only if $\mathbf{v}$ and $ID$ are conditionally independent given $S$.*

Theorem 1 demonstrates that the cascaded modeling consistently produces a joint distribution with lower information entropy. This implies a more compact and informative representation that facilitates model learning. The detailed proof is provided in Appendix F.

## 4 Experiment

This section presents a comprehensive evaluation of the COBRA framework using both public and industrial datasets. Our experiments focus on assessing COBRA's ability to improve recommendation accuracy and diversity, while also validating its practical effectiveness through offline and online evaluations.

### 4.1 Public Dataset Experiments

**Datasets.** In our experiments, we evaluate the performance of COBRA using the Amazon Product Reviews dataset [35, 36]. Our analysis focuses on three specific subsets: "Beauty," "Sports and Outdoors," and "Toys and Games." For each subset, we construct item embeddings leveraging attributes such as title, price, category, and description. To ensure data quality, we apply a 5-core filtering process, eliminating items with fewer than five user interactions and users with fewer than five item interactions. Detailed statistics of the datasets are presented in Appendix C.1.

**Evaluation Metrics.** For the evaluation of recommendation accuracy and ranking quality, we employ Recall@K and NDCG@K, specifically at $K = 5$ and $K = 10$. These metrics provide insights into the system's ability to accurately recommend relevant items and maintain a high-quality ranking order.

**Implementation Details.** In our approach, we adopt a method for generating semantic IDs similar to the one used in [19]. However, unlike [19], which uses a different configuration, we employ a 3-level semantic ID structure, where each level corresponds to a codebook size of 32. These semantic IDs are generated using the T5 model. COBRA is implemented with a lightweight architecture, featuring a 1-layer encoder and a 2-layer decoder.

**Baselines.** To comprehensively evaluate the performance of our proposed COBRA method, we compare it with the following recommendation methods (which are described briefly in Appendix B): P5 [33], Caser [28], HGN [37], GRU4Rec [27], SASRec [8], FDSA [38], BERT4Rec [9], S³-Rec [39], and TIGER [19].

**Results.** COBRA consistently surpasses all baseline models across various metrics, as presented in Table 1. On the "Beauty" dataset, COBRA achieves a Recall@5 of 0.0537 and a Recall@10 of 0.0725, exceeding the previous strongest baseline model (TIGER) by 18.3% and 11.9%, respectively. For the "Sports and Outdoors" dataset, COBRA records a Recall@5 of 0.0305 and an NDCG@5 of 0.0215, outperforming TIGER by 15.5% and 18.8%, respectively. On the "Toys and Games" dataset, COBRA attains a Recall@10 of 0.0462 and an NDCG@10 of 0.0515, surpassing TIGER by 24.5% and 19.2%, respectively.

Table 1: Performance comparison on public datasets. The best metric for each dataset is highlighted in bold, while the second-best is underlined.

| | Method | R@5 | N@5 | R@10 | N@10 |
|---|---|---|---|---|---|
| Beauty | P5 | 0.0163 | 0.0107 | 0.0254 | 0.0136 |
| | Caser | 0.0205 | 0.0131 | 0.0347 | 0.0176 |
| | HGN | 0.0325 | 0.0206 | 0.0512 | 0.0266 |
| | GRU4Rec | 0.0164 | 0.0099 | 0.0283 | 0.0137 |
| | BERT4Rec | 0.0203 | 0.0124 | 0.0347 | 0.0170 |
| | FDSA | 0.0267 | 0.0163 | 0.0407 | 0.0208 |
| | SASRec | 0.0387 | 0.0249 | 0.0605 | 0.0318 |
| | $S^3$-Rec | 0.0387 | 0.0244 | 0.0647 | 0.0327 |
| | TIGER | 0.0454 | 0.0321 | 0.0648 | 0.0384 |
| | **COBRA** | **0.0537** | **0.0395** | **0.0725** | **0.0456** |
| Sports | P5 | 0.0061 | 0.0041 | 0.0095 | 0.0052 |
| | Caser | 0.0116 | 0.0072 | 0.0194 | 0.0097 |
| | HGN | 0.0189 | 0.0120 | 0.0313 | 0.0159 |
| | GRU4Rec | 0.0129 | 0.0086 | 0.0204 | 0.0110 |
| | BERT4Rec | 0.0115 | 0.0075 | 0.0191 | 0.0099 |
| | FDSA | 0.0182 | 0.0122 | 0.0288 | 0.0156 |
| | SASRec | 0.0233 | 0.0154 | 0.0350 | 0.0192 |
| | $S^3$-Rec | 0.0251 | 0.0161 | 0.0385 | 0.0204 |
| | TIGER | 0.0264 | 0.0181 | 0.0400 | 0.0225 |
| | **COBRA** | **0.0305** | **0.0215** | **0.0434** | **0.0257** |
| Toys | P5 | 0.0070 | 0.0050 | 0.0121 | 0.0066 |
| | Caser | 0.0166 | 0.0107 | 0.0270 | 0.0141 |
| | HGN | 0.0321 | 0.0221 | 0.0497 | 0.0277 |
| | GRU4Rec | 0.0097 | 0.0059 | 0.0176 | 0.0084 |
| | BERT4Rec | 0.0116 | 0.0071 | 0.0203 | 0.0099 |
| | FDSA | 0.0228 | 0.0140 | 0.0381 | 0.0189 |
| | SASRec | 0.0463 | 0.0306 | 0.0675 | 0.0374 |
| | $S^3$-Rec | 0.0443 | 0.0294 | 0.0700 | 0.0376 |
| | TIGER | 0.0521 | 0.0371 | 0.0712 | 0.0432 |
| | **COBRA** | **0.0619** | **0.0462** | **0.0781** | **0.0515** |

**Ablation Study.** To validate the necessity of COBRA's key components and understand their individual contributions, we compare the full model against three variants. *COBRA w/o ID* removes sparse IDs, relying solely on dense vectors. *COBRA w/o Dense* removes dense vectors, using only sparse IDs. *COBRA w/o BeamFusion* removes the BeamFusion module during

Table 2: Ablation study on public datasets (Recall@10).

| Method | Beauty | Sports | Toys |
|---|---|---|---|
| **COBRA** | **0.0725** | **0.0434** | **0.0781** |
| COBRA w/o Dense | 0.0656 | 0.0331 | 0.0713 |
| COBRA w/o ID | 0.0681 | 0.0365 | 0.0653 |
| COBRA w/o BeamFusion | 0.0714 | 0.0413 | 0.0769 |

inference, using top-1 sparse ID and nearest-neighbor retrieval for top-$k$ results. As shown in Table 2, the removal of any key component leads to a consistent performance drop. *COBRA w/o Dense* shows a significant decline, highlighting the limitations of using only discrete sparse IDs, which fail to capture fine-grained semantic nuances. *COBRA w/o ID* also underperforms, demonstrating the importance of sparse IDs in offering a structural framework that supports coarse-to-fine generation. *COBRA w/o BeamFusion* also exhibits a performance drop.

## 4.2 Industrial-scale Experiments

**Dataset.** To comprehensively evaluate the proposed COBRA method, we utilize a large-scale industrial dataset from a major information feed platform, which contains 5 million users and 2 million advertisements across diverse recommendation scenarios. Advertisements are represented via attributes such as title, industry labels, brand, and campaign text, encoded into two-level sparse IDs

Table 3: Performance comparison on industrial dataset

| Method | R@50 | R@100 | R@200 | R@500 | R@800 |
|---|---|---|---|---|---|
| **COBRA** | **0.1180** | **0.1737** | **0.2470** | **0.3716** | **0.4466** |
| COBRA w/o ID | 0.0611 | 0.0964 | 0.1474 | 0.2466 | 0.3111 |
| COBRA w/o Dense | 0.0690 | 0.1032 | 0.1738 | 0.2709 | 0.3273 |
| COBRA w/o BeamFusion | 0.0856 | 0.1254 | 0.1732 | 0.2455 | 0.2855 |

and dense vectors to capture multi-granularity semantic information. A more detailed description of the dataset can be found in Appendix C.2.

**Evaluation Metrics.** For offline evaluation, we employ Recall@K as the evaluation metric, testing with $K \in \{50, 100, 200, 500, 800\}$. This metric provides a measure of the model's ability to accurately retrieve relevant recommendations at various thresholds.

**Implementation Details.** COBRA is built upon a Transformer-based architecture. In this framework, the text encoder processes advertisement text into sequences, which are then handled by the sparse ID head to predict 2-level semantic IDs configured as $32 \times 32$.

**Results.** For further analysis on the industrial dataset, we also compare COBRA against the variants defined in the previous section, i.e., *COBRA w/o ID*, *w/o Dense*, and *w/o Beam-Fusion*. Notably, the *COBRA w/o Dense* variant employs finer-grained 3-level semantic IDs ($256 \times 256 \times 256$) to ensure its sufficient fine-grained modeling capacity, compensating for its lack of dense vectors. As shown in Table 3, COBRA consistently outperforms all its variants across all evaluated metrics. At $K = 500$, COBRA achieves a Recall@500 of 0.3716, representing a 42.2% improvement over the CO-BRA w/o Dense variant. When $K = 800$, CO-BRA attains a Recall@800 of 0.4466, reflecting a 43.6% improvement over the COBRA w/o ID

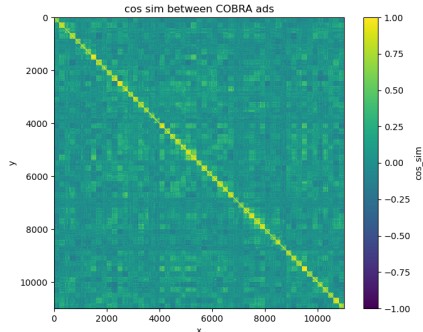

Figure 4: Cosine Similarity Matrix (full comparison in Appendix 6)

variant and a 36.1% enhancement compared to *COBRA w/o BeamFusion*. At relatively smaller values of $K$, the absence of dense or ID representations results in more pronounced performance declines, underscoring the importance of cascaded representations for achieving granularity and precision. Conversely, as the recall size $K$ increases, the performance advantages associated with BeamFusion become increasingly evident, demonstrating its effectiveness in practical industrial recall systems. The results further underscore the contributions of specific components: Excluding sparse IDs leads to a recall reduction ranging from 26.7% to 41.5%, highlighting the critical role of semantic categorization. The removal of dense vector results in a performance drop between 30.3% and 48.3%, underscoring the importance of fine-grained modeling. Eliminating BeamFusion results in a recall decrease of 27.5% to 36.1%, emphasizing the significance of fusion strategy.

### 4.3 Further Analysis

**Analysis of Representation Learning.** The heatmap in Figure 4 demonstrates COBRA's strong intra-ID cohesion and inter-ID separation, indicating effective capture of both item-specific features and categorical semantics. Quantitative verification through difference analysis is provided in Appendix D. Further validation of COBRA's embeddings is achieved through visualizing the distribution of advertisement embeddings in a two-dimensional space using t-SNE. By randomly sampling 10,000 advertisements, distinct clustering centers for various categories are observed. Figure 5a reveals that advertisements are effectively clustered by category, indicating strong cohesion within categories. The clusters in purple, teal, light green, and dark green correspond primarily to advertisements for novels, games, legal services, and clothing, respectively. This demonstrates that the advertisement representations effectively capture semantic information.

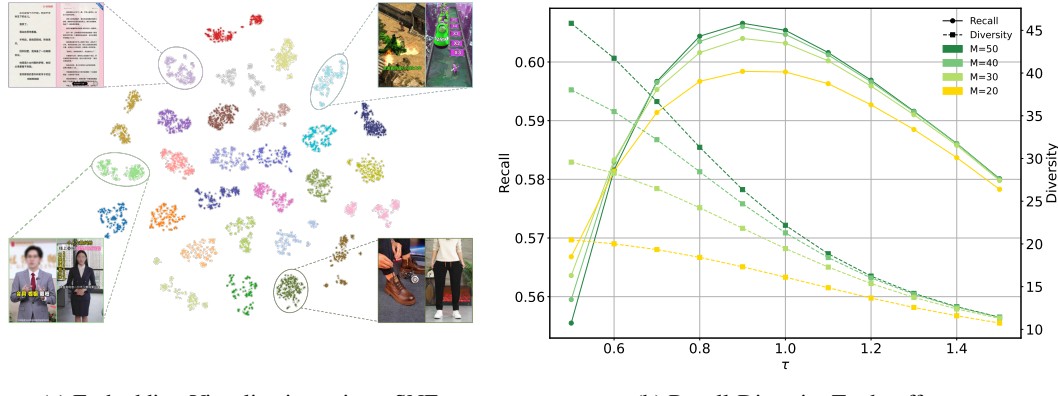

(a) Embedding Visualization using t-SNE    (b) Recall-Diversity Trade-off

Figure 5: (a) t-SNE visualization of 10,000 randomly sampled advertisement embeddings. (b) Recall@2000 and diversity metrics under different $\tau$ values.

**Recall-Diversity Equilibrium.** To analyze the trade-off between accuracy and diversity in COBRA, we examine recall-diversity curves, which depict how Recall@2000 and diversity metrics evolve with the coefficient $\tau$ in the BeamFusion mechanism, while keeping $\phi$ fixed. As depicted in Figure 5b, increasing $\tau$ generally leads to a decrease in diversity. COBRA achieves an optimal balance between recall and diversity at $\tau = 0.9$. At this point, the model maintains high accuracy while ensuring a sufficiently diverse set of retrieved items. Specifically, when $M = 50$, compared to $\tau = 1.0$, setting $\tau = 0.5$ results in a 4.99% decrease in recall, but brings more than double the diversity. Meanwhile, $\tau = 0.9$ leads to a 0.12% increase in recall and an 18.80% relative improvement in diversity. This fine-grained control over $\tau$ and $\phi$ allows for adjusting the emphasis on accuracy or diversity according to specific business objectives. Platforms prioritizing exploration can reduce $\tau$ to enhance diversity. This flexibility distinguishes COBRA from models with fixed retrieval strategies, making it adaptable to various recommendation scenarios.

**Online Results.** To validate COBRA's real-world effectiveness, we conducted online A/B tests on a major information feed platform. The experiment covered 10% of user traffic, ensuring statistical significance. The primary evaluation metrics were conversion and Average Revenue Per User (ARPU), which directly reflect user engagement and economic value. Within the traffic segment exposed to our proposed strategy, COBRA achieved a 3.60% increase in conversion and a 4.15% increase in ARPU. These results demonstrate that COBRA's hybrid architecture not only improves recommendation quality in offline evaluations but also drives measurable business outcomes in production environments.

## 5 Conclusion

In this work, we introduced COBRA, a generative recommendation framework that combines sparse and dense representations for enhanced accuracy and diversity. COBRA employs a coarse-to-fine generation process, starting with a sparse ID to capture the categorical essence of an item and refining it with a dense vector. Our extensive experiments on public and industrial datasets demonstrate that COBRA achieves superior performance over state-of-the-art methods, delivering high accuracy with controllable diversity. These gains are further validated through online A/B tests, confirming the method's practical applicability. In the future, we intend to incorporate more multi-domain and multi-modal information to further enhance our framework's effectiveness. Additionally, we will explore performance optimizations in the generative approach to improve its efficiency and scalability.

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

# A Related Work

**Sequential Dense Recommendation.** Sequential dense recommendation utilize user interaction histories to learn representations for personalization [8, 39, 29], capturing both long-term and short-term preferences [40–43]. GRU4Rec [27] first applied RNNs [44] to model sequential behavior, effectively handling temporal dependencies. Caser [28] adopted CNNs [45] to extract local sequential patterns from behavior matrices. Transformer-based models, such as SASRec [8] and BERT4Rec [9], leverage self-attention to model complex user behavior, with SASRec focusing on autoregressive predictions and BERT4Rec adopting bidirectional context encoding. Advanced architectures like PinnerFormer [46] and FDSA [38] further enhance user modeling by integrating multi-source features and capturing long-range dependencies. Recent efforts (e.g., ZESRec [30], UniSRec [31], RecFormer [29]) have moved towards cross-domain recommendation and richer feature integration, often via contrastive learning and unified language-sequence modeling.

**Generative Recommendation.** With the emergence of large generative models, recommendation is increasingly framed as a generation task [32, 47–51]. P5 [33] reformulates various recommendation tasks as language generation, enabling unified modeling and prompting strategies. TIGER [19] applies residual quantized autoencoders to produce semantic item identifiers, which transformers then generate from user sequences. LC-Rec [21] aligns these semantic IDs with collaborative signals, while IDGenRec [23] leverages large language models for unique, dense textual identifiers. SEATER [34] and ColaRec [22] focus on aligning semantic and collaborative spaces or maintaining semantic consistency via structured indexes. Despite their advantages, generative models relying on discrete IDs may lose fine-grained preference information [52], and natural language generation may be less aligned with recommendation-specific signals [53]. Hybrid approaches like LIGER [26] jointly generate sparse IDs and dense vectors, narrowing the gap between retrieval paradigms. Nevertheless, challenges remain in achieving optimal flexibility and representation granularity.

# B Baselines

To rigorously evaluate the effectiveness of our proposed COBRA method, we benchmark it against a range of recommendation methods.

- **P5** [33]: This innovative approach transforms recommendation tasks into natural language sequences, utilizing the capabilities of language models to unify various recommendation scenarios.

- **Caser** [28]: Caser employs convolutional layers to effectively capture sequential user behavior patterns, modeling high-order Markov Chains.

- **HGN** [37]: The Hierarchical Gating Network(HGN) is designed to capture both long-term and short-term user interests through a sophisticated gating architecture, facilitating personalized recommendations.

- **GRU4Rec** [27]: As a pioneering RNN-based approach, GRU4Rec leverages gated recurrent units to model user behaviors in sequential recommendation tasks, paving the way for subsequent developments in the field.

- **SASRec** [8]: As a Transformer-based model, SASRec focuses on long-term dependencies in user interactions, employing self-attention mechanisms for precise sequential recommendations.

- **FDSA** [38]: By integrating item features with embeddings, the Feature-level Deeper Self-Attention(FDSA) Network enriches the input sequence, leveraging self-attentive mechanisms to enhance recommendation quality.

- **BERT4Rec** [9]: Utilizing a bidirectional self-attention framework with a cloze task objective, BERT4Rec overcomes the limitations of traditional uni-directional models, offering robust recommendation capabilities.

- **S³-Rec** [39]: S³-Rec leverages contrastive learning to bolster the recommendation process, employing self-supervised techniques to enhance sequential recommendation performance.

- **TIGER** [19]: TIGER employs RQ-VAE for encoding item content features and leverages a Transformer for generative retrieval, showcasing a novel approach to incorporating content features in recommendation tasks.

These methods collectively represent the forefront of recommendation technology, embodying diverse methodologies from sequential and dense recommendation to generative approaches.

## C  Dataset Statistics

### C.1  Public Datasets.

Table 4: Summary of dataset statistics for three real-world benchmarks.

| Dataset | Users | Items | Avg. Length | Med. Length |
|---|---|---|---|---|
| Beauty | 22,363 | 12,101 | 8.87 | 6 |
| Sports and Outdoors | 35,598 | 18,357 | 8.32 | 6 |
| Toys and Games | 19,412 | 11,924 | 8.63 | 6 |

Our study utilizes the Amazon Product Reviews dataset [35, 36], which spans user reviews and product information from May 1996 to July 2014. To comprehensively explore the effectiveness of recommendation methods, we selected three categories: "Beauty," "Sports and Outdoors," and "Toys and Games." Table 4 provides a concise summary of these datasets. During data preprocessing, we constructed users' historical item interaction sequences based on review timestamps, excluding users with fewer than five reviews. For evaluation, we adopted the widely-used leave-one-out strategy: the last item in each user's sequence served as the test sample, the second-to-last as the validation sample, and the remaining items as training data. The "Beauty" dataset contains 22,363 users and 12,101 items, featuring an average sequence length of approximately 8.87, with a median of 6. The "Sports and Outdoors" dataset comprises 35,598 users and 18,357 items, with an average sequence length of 8.32 and a median of 6. Similarly, the "Toys and Games" dataset includes 19,412 users and 11,924 items, with an average sequence length of about 8.63 and a median of 6.

### C.2  Industrial Dataset Details.

To thoroughly evaluate the proposed COBRA method, we employ a large-scale industrial dataset derived from user interaction logs on a major information feed platform. This dataset covers multiple recommendation scenarios, including list-page, dual-column, and short-video recommendations, and contains approximately 5 million users and 2 million advertisements, providing a comprehensive reflection of real-world user behaviors and advertising content.

Advertisers and advertisements are described by attributes such as title, industry labels, brand, and campaign text. To effectively capture both coarse-grained and fine-grained semantic information, these attributes are encoded into two-level sparse IDs alongside dense vector representations. This dual encoding enables COBRA to model user preferences and item characteristics more accurately.

The dataset is divided into two parts: the training set $D_{\text{train}}$ and the test set $D_{\text{test}}$. The training set consists of user interaction logs collected over the first 60 days, covering recommendation content and user behaviors during this period. The test set is constructed from logs recorded on the day immediately following the training period and serves as a benchmark for model performance evaluation. This chronological split ensures the temporal consistency of training and testing processes, improving the reliability of the evaluation.

## D  Supplementary Similarity Analysis

The COBRA model exhibits significant intra-ID cohesion and inter-ID separation, as demonstrated in the top heatmap of Figure 4. This suggests that COBRA's dense embeddings proficiently capture detailed item characteristics while preserving semantic consistency within categories. Conversely, the model variant without sparse IDs (Figure 6a) shows weaker category separation, underscoring the importance of sparse IDs in maintaining semantic structure. The difference matrix in Figure 6b quantitatively confirms that incorporating sparse IDs enhances both cohesion and separation.

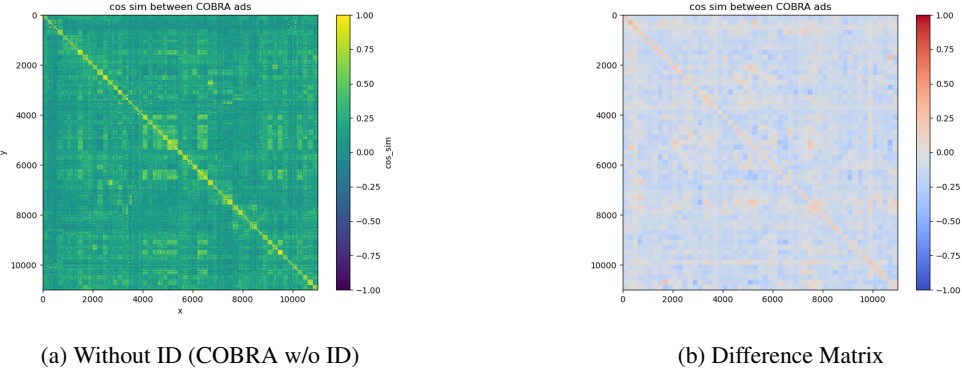

(a) Without ID (COBRA w/o ID)                    (b) Difference Matrix

Figure 6: Complete similarity matrix comparison: (a) Weaker separation without ID, (b) Quantitative improvement from sparse IDs

# E    Pseudocode for Coarse-to-Fine Generation

---
**Algorithm 1** Inference with BeamFusion
---
1: **Input**: input_seq $\mathbf{S}_{1:T}$, beam_size $M$, nn_num $N$, recall_num $K$, candidate items $\mathbf{a}$
2: **Output**: Top $K$ recommendations $\mathcal{R}$
3:
4: **procedure** FORWARD($\cdots$)
5:     **Sparse ID Generation**:
6:     **for** each ID hierarchy **do**
7:         Run transformer forward pass
8:         Compute ID logits and scores $\{\hat{\mathbf{ID}}_{T+1}^{k}\}_{k=1}^{M}$
9:         Update beam scores and decoder input
10:    **end for**
11:    **Dense Vector Refinement**:
12:        Obtain final decoder output $\hat{\mathbf{v}}_{T+1}^{k}$
13:        Compute similarity scores $\cos(\hat{\mathbf{v}}_{T+1}^{k}, \mathbf{a})$
14:        Filter logits using generated ID
15:    **BeamFusion Mechanism**:
16:        Combine beam scores $\phi_{\hat{\mathbf{ID}}_{T+1}^{k}}$ with similarity scores $\cos(\hat{\mathbf{v}}_{T+1}^{k}, \mathbf{a})$
17:        Select top $K$ candidates based on fused scores
18:    **Prepare Results**:
19:        Collect final ID sequences, retrieved items, and scores
20: **end procedure**
---

# F    Proof of Theorem 1

We calculate the information entropy (for discrete variable $ID$) and differential entropy (for continuous variable $\mathbf{v}$) under both probability distributions given by Eq. 9 and Eq. 10. We denote the discrete entropy by $H(\cdot)$ and the differential entropy by $h(\cdot)$.

**Independent Modeling (Eq. 9):**

$$
\begin{aligned}
H_{indep} &= -\mathbb{E}_{P_{indep}}[\log P(ID, \mathbf{v}|S)] \\
&= -\mathbb{E}_{P_{indep}}[\log(P(ID|S) \cdot P(\mathbf{v}|S))] \\
&= -\mathbb{E}_{P_{indep}}[\log P(ID|S)] - \mathbb{E}_{P_{indep}}[\log P(\mathbf{v}|S)] \\
&= H(ID|S) + h(\mathbf{v}|S)
\end{aligned}
\tag{12}
$$

Hence, under independent modeling, the total entropy decomposes additively over $ID$ and $\mathbf{v}$.

**Cascaded Modeling (Eq. 10):**

$$\begin{aligned}
H_{cascaded} &= -\mathbb{E}_{P_{cascaded}}[\log P(ID, \mathbf{v}|S)] \\
&= -\mathbb{E}_{P_{cascaded}}[\log(P(ID|S) \cdot P(\mathbf{v}|ID, S))] \\
&= -\mathbb{E}_{P_{cascaded}}[\log P(ID|S)] - \mathbb{E}_{P_{cascaded}}[\log P(\mathbf{v}|ID, S)] \\
&= H(ID|S) + h(\mathbf{v}|ID, S)
\end{aligned} \tag{13}$$

Hence, cascaded modeling captures the full joint uncertainty between $ID$ and $\mathbf{v}$.

**Comparison:** By the property of conditional entropy [54]:

$$h(\mathbf{v}|S) \geq h(\mathbf{v}|ID, S) \tag{14}$$

Equality holds if and only if $\mathbf{v}$ and $ID$ are conditionally independent given $S$.

Substituting this inequality into Eq. 12 and 13 yields:

$$H_{indep} \geq H_{cascaded} \tag{15}$$

which confirms Theorem 1.

## G  Computational Cost Analysis

We define $L$ as the sequence length, $T$ as the number of tokens per item and $D$ as the embedding dimension.

Table 5: Training Complexity. (Please add citations for TIGER/SASRec)

| Component | COBRA | TIGER [19] | SASRec [8] |
|---|---|---|---|
| Item Feature Encoder | $\mathcal{O}(L \cdot T^2 \cdot D + L \cdot T \cdot D^2)$ | N/A | N/A |
| Sequential Model | $\mathcal{O}(L^2 \cdot D + L \cdot D^2)$ | $\mathcal{O}(L^2 \cdot D + L \cdot D^2)$ | $\mathcal{O}(L^2 \cdot D + L \cdot D^2)$ |

Table 6: Inference Complexity. (Please add citations for TIGER/SASRec)

| Component | COBRA | TIGER [19] | SASRec [8] |
|---|---|---|---|
| Item Feature Encoder | $\mathcal{O}(1)$ (cached) | N/A | N/A |
| Sequential Model | $\mathcal{O}(L^2 \cdot D + L \cdot D^2)$ | $\mathcal{O}(L^2 \cdot D + L \cdot D^2)$ | $\mathcal{O}(L^2 \cdot D + L \cdot D^2)$ |

To ensure practical efficiency, we employ several key techniques during implementation. These include sequence Packing and FlashAttention to minimize computational waste. Furthermore, encoder caching is utilized to decouple the encoder computation, significantly speeding up the inference process. Thanks to these optimizations, COBRA achieves over 30% Model FLOPs Utilization (MFU), and has been successfully deployed, serving over 200 million daily users.

