# OpenReview forum: "Sparse Meets Dense: Unified Generative Recommendations with Cascaded Sparse-Dense Representations"
_NeurIPS.cc/2025/Conference — NeurIPS 2025 poster_

### Official Review · Reviewer_7Wfj · 2025-06-29

**Clarity:** 2
**Significance:** 3
**Originality:** 2
**Rating:** 4
**Confidence:** 5

**Summary:**

This paper introduces COBRA, a generative recommendation framework that combines sparse semantic IDs and dense item vectors through a cascaded generation process. By leveraging a coarse-to-fine strategy and a novel BeamFusion inference method, COBRA improves both accuracy and diversity. Extensive evaluations on public and industrial datasets, including online A/B tests, demonstrate its effectiveness over strong baselines.

**Questions:**

1. Why is there such a large performance gap between GRU4Rec and SASRec across the three datasets?
2. Why didn’t the authors include comparisons with the most recent methods in both traditional sequential recommendation and generative recommendation?
3. I suggest that the authors revise the methodology section using more standard equation formatting practices.
4. Although the paper presents corresponding experimental results, it fails to clearly illustrate the limitations of TIGER and explain why integrating sparse semantic IDs and dense vectors can effectively address the issues in TIGER.

**Ethical Concerns:**

["NO or VERY MINOR ethics concerns only"]

**Final Justification:**

Thanks author's response. I will maintain my original positive score.

**Limitations:**

Yes

**Quality:**

3

**Strengths And Weaknesses:**

Strengths:
1. The paper presents a well-motivated and novel cascaded representation learning strategy that tightly integrates sparse semantic IDs and dense vectors in both training and inference, addressing limitations of previous generative retrieval methods that rely solely on one modality.
2. The model is not only validated through rigorous offline experiments but also deployed in a real-world recommendation platform.
The authors provide extensive quantitative evaluations, including ablations and diversity trade-off analysis, offering strong evidence for the effectiveness and robustness of their approach.

Weaknesses:
1. As far as I know, the only difference between GRU4Rec and SASRec is that the former uses RNN for sequential modeling, while the latter adopts attention mechanisms. While attention-based methods may have certain advantages, why is there such a significant performance gap between the two on all three datasets?
I believe the baselines used in the experiments can be roughly divided into two categories: traditional sequential recommendation methods, such as GRU4Rec, and generative approaches, such as P5 and TIGER. However, the authors did not compare their method with the latest baselines from either category.
2. The authors did not present each equation on a separate line with a corresponding number in the methodology section, which slightly deviates from standard academic writing conventions.
3. While the empirical results are convincing, the paper lacks a formal theoretical analysis of why cascaded sparse-dense generation yields superior performance or how error propagation between stages is mitigated.

---

> ### Author Rebuttal · Authors · 2025-07-30
>
> We sincerely thank the reviewer for the constructive feedback and thoughtful questions. Below, we address the key concerns raised.
>
> ---
>
> ### **For Weakness1 & Question 1: Large performance gap between GRU4Rec and SASRec**
>
> We appreciate the reviewer’s attention to this detail. Indeed, both GRU4Rec[1] and SASRec[2] are classical sequential recommendation models, but they differ significantly in their capacity to capture long-range dependencies and sequence context.
>
> GRU4Rec leverages RNNs, which are known to struggle with modeling long user behavior sequences due to vanishing gradient issues. In contrast, SASRec employs self-attention mechanisms that allow the model to directly attend to any previous item in the sequence, thereby capturing richer patterns and long-range dependencies more effectively. This architectural advantage explains SASRec’s consistent outperformance on all datasets. This observation is also consistently reported in other research, such as TIGER[3] and S$^3$-Rec[4].
>
> ---
>
> ### **For Weakness 1 & Question 2: Comparisons with the most recent methods**
>
> COBRA's core contribution lies in its novel cascaded sparse-dense representation for items, which integrates the strengths of both generative and dense retrieval paradigms through a coarse-to-fine generation process. Our primary focus is on this paradigm shift. We contend that most existing generative recommendation improvements focus on optimizing sparse ID generation, which is orthogonal to our paradigm-level innovation. We believe that such improvements could potentially be integrated into COBRA's sparse ID generation.
>
> To further validate COBRA's effectiveness, we conducted additional experiments on the Amazon Beauty dataset, comparing COBRA with recent representative approaches: LC-Rec[5] , GenRec[6] , and LETTER[7]. The results are as follows:
>
> | Method         | Recall@5 | Recall@10 | NDCG@5  | NDCG@10 |
> | :------------- | :------- | :-------- | :------ | :------ |
> | LC-Rec          | 0.0482   | 0.0681    | 0.0327  | 0.0409  |
> | LETTER           | 0.0505   | 0.0703    | 0.0355  | 0.0418  |
> | GenRec          | 0.0515   | 0.0641    | 0.0397  | 0.0439  |
> | COBRA (Ours)    | 0.0537   | 0.0725    | 0.0395  | 0.0456  |
>
> As the results demonstrate, COBRA consistently achieves strong performance. Specifically, it shows a +4.3% improvement in Recall@5 compared to GenRec , a +13.1% improvement in Recall@10 over LC-Rec , and an +8.7% improvement in NDCG@10 compared to GenRec. While GenRec shows a slightly higher NDCG@5, COBRA generally achieves superior holistic performance, confirming the unique value of our cascaded sparse-dense paradigm.
>
> ---
>
> ### **For Weakness 2 & Question 3: Equation format**
>
> We appreciate the reviewer’s suggestion regarding presentation. To improve clarity and align with academic writing standards, we will revise the methodology section by separating each key equation into its own line with proper numbering. This will be reflected in the final submission.
>
> ---
>
> ### **For Weakness 3 & Question 4: Formal theoretical analysis on sparse-dense cascading**
>
> The reviewer raises an excellent point regarding the theoretical analysis of our cascaded design. Our approach addresses a key limitation of ID-based models like TIGER  where discrete semantic IDs can limit expressiveness for fine-grained user intent. We augment each item with a dense representation to capture more comprehensive information, thus each item is represented by a (sparse, dense) pair. We theoretically demonstrate the superiority of our cascaded prediction modeling compared to an independent prediction approach.
>
> Let $S_{1:t}$ be a user's behavior sequence. For an item, $C$ represents a discrete random variable for its sparse ID, and $X$ represents a continuous random variable for its dense vector. We denote the information entropy of a discrete random variable as $H$ and the differential entropy of a continuous random variable as $h$. Our objective is to model $P(Item\_{t+1} | S\_{1:t})$ for predicting the next interacted item. We consider two distinct approaches for modeling $Item_{t+1}$:
>
> **Approach 1: Independent Prediction** Predict $C$ and $X$ separately based on the user's historical sequence $S\_{1:t}$. The probability distribution for this approach is formulated as:
>     $$P(Item_{t+1} | S_{1:t}) = P(C | S_{1:t}) \times P(X | S_{1:t}) \quad (1)$$
>
> **Approach 2: Cascaded Prediction** First predict the sparse information of the $Item_{t+1}$ using the user's historical sequence $S\_{1:t}$, and then utilize this sparse information along with the preceding sequence $S\_{1:t}$ to predict the dense vector of the next item. The probability distribution for this approach is formulated as:
>     $$P(Item_{t+1} | S_{1:t}) = P(C | S_{1:t}) \times P(X | C, S_{1:t}) = P(C, X | S_{1:t}) \quad (2)$$
>
> We demonstrate that the cascaded prediction modeling (Approach 2) consistently yields superior performance compared to independent prediction modeling.
>
> **Theorem 1: Superiority of Cascaded Modeling**
> Let $E_1$ be the information entropy of the independent prediction modeling probability distribution (Equation 1), and $E_2$ be the information entropy of the cascaded prediction modeling probability distribution (Equation 2). Then, it always holds that $E_2 \leq E_1$, where equality holds if and only if the random variables $C$ and $X$ are conditionally independent given $S\_{1:t}$.
>
> **Proof:**
> We calculate the information entropy for both probability distributions given by equations (1) and (2).
>
> For independent prediction modeling (Equation 1):
> $$
> \begin{aligned}
> E\_1 &= -\sum\_{c} \int_{x} P(c|S_{1:t}) \times P(x | S_{1:t}) \log(P(c|S_{1:t}) \times P(x | S_{1:t})) dx \\\\
> &= -\sum_{c} \int_{x} P(c|S_{1:t}) \times P(x | S_{1:t}) \log(P(c|S_{1:t})) dx \\\\
> &\quad -\sum_{c} \int_{x} P(c|S_{1:t}) \times P(x | S_{1:t}) \log(P(x | S_{1:t})) dx \\\\
> &= H(C | S_{1:t}) + h(X | S_{1:t})
> \end{aligned}
> $$
>
> Hence, under independent modeling, the total entropy decomposes additively over $C$ and $X$.
>
> For cascaded prediction modeling (Equation 2):
> $$
> \begin{aligned}
> E_2 &= -\sum_{c} \int_{x} P(c|S_{1:t}) \times P(x | c, S_{1:t}) \log(P(c|S_{1:t}) \times P(x | c, S_{1:t})) dx \\\\
> &= -\sum_{c} \int_{x} P(c, x|S_{1:t}) \log(P(c, x|S_{1:t})) dx \\\\
> &= H(C, X | S_{1:t})
> \end{aligned}
> $$
>
> Hence, cascaded modeling captures the full joint uncertainty of $C$ and $X$. By the subadditivity of information entropy[3]:$$E_2 = H(C, X | S_{1:t}) \leq H(C | S_{1:t}) + h(X| S_{1:t}) = E_1$$
> Equality holds if and only if random variables $C$ and $X$ are conditionally independent.
>
> Theorem 1 demonstrates that our cascaded modeling scheme always yields a probability distribution with lower information entropy. This implies that the resulting probability distribution is more concentrated, which is beneficial for model learning. Equality holds only when random variables $C$ and $X$ are conditionally independent, in which case $P(x|c, S_{1:t}) = P(x | S_{1:t})$. From equations (1) and (2), it is evident that both modeling approaches are equivalent under this condition. Consequently, cascaded modeling is generally superior to independent modeling.
>
> ---
>
> ### Conclusion
>
> We thank the reviewer again for their detailed and insightful comments. We will incorporate all of the above improvements into the final version to enhance clarity, completeness, and academic rigor.
>
> ---
>
> ### Reference
> 1. Hidasi et al., "Session-based Recommendations with Recurrent Neural Networks"
> 2. Kang & McAuley, "Self-Attentive Sequential Recommendation", ICDM 2018
> 3. Rajput et al., "Recommender Systems with Generative Retrieval", NeurIPS 2023
> 4. Zhou et al., "S^3-Rec: Self-Supervised Learning for Sequential Recommendation with Mutual Information Maximization", CIKM 2020
> 5. Zheng et al., "Adapting Large Language Models by Integrating Collaborative Semantics for Recommendation", ICDE 2024
> 6. Cao & Lio, "GenRec: Generative Sequential Recommendation with Large Language Models", ECIR 2024
> 7. Wang et al., "Learnable Item Tokenization for Generative Recommendation", CIKM 2024

---

### Official Review · Reviewer_XoPo · 2025-07-01

**Clarity:** 3
**Significance:** 2
**Originality:** 2
**Rating:** 4
**Confidence:** 4

**Summary:**

This paper presents COBRA (Cascaded Organized Bi-Represented generAtive retrieval), a unified framework for generative recommendation that integrates both sparse semantic IDs and dense vector representations in a cascaded manner. The model takes a user’s interaction history encoded as a sequence of sparse-dense pairs and uses a Transformer decoder to alternately predict the next item’s sparse ID followed by its dense vector. The dense representations are trained end-to-end using contrastive learning, and the inference adopts a coarse-to-fine generation strategy supported by a BeamFusion module for diversity control. Experiments on standard benchmarks show COBRA outperforms previous state-of-the-art generative recommenders in both accuracy and diversity.

**Questions:**

See above weakness

**Ethical Concerns:**

["NO or VERY MINOR ethics concerns only"]

**Final Justification:**

The authors addressed some of my concerns in the rebuttal and demonstrated effectiveness in industrial scenarios, which led me to slightly raise my score. However, this paper is still largely engineering-focused with limited novelty. Moreover, in the rebuttal’s proof for “Weakness 3,” the authors claim that lower entropy facilitates learning, but in recommender systems, considering the need for diversity, lower entropy may not necessarily be beneficial. In addition, when addressing the scaling issue, the authors did not provide concrete improvements in real-world scenarios, which reduces the credibility of their claims.

**Limitations:**

yes

**Quality:**

2

**Strengths And Weaknesses:**

Strengths

1. The use of sparse IDs as coarse-grained anchors helps reduce the learning difficulty of dense representations and improves controllability in inference.

2. The introduction of BeamFusion adds flexibility and diversity to the generative recommendation process.

3. Empirical results on public benchmarks demonstrate strong performance gains over existing generative recommendation models such as TIGER.


Weaknesses

1. The baselines primarily include sparse ID-based models (e.g., TIGER), but miss stronger or more comparable systems like Learnable Item Tokenization for Generative Recommendation, which also explore hybrid sparse-dense paradigms. The absence of such baselines makes it difficult to fully assess COBRA's relative effectiveness.

2. The model architecture used in experiments is relatively small (1-layer encoder and 2-layer decoder), which raises concerns about scalability. Recent models like OneRec suggest that more expressive architectures are becoming standard, and it would strengthen the work to evaluate COBRA’s performance under larger-scale settings.

3. The proposed method lacks sufficient theoretical analysis to justify why the cascaded structure improves performance. While empirical results are promising, scenarios like cold-start—where sparse IDs may be poorly learned—are not thoroughly discussed. The assumption that sparse IDs always enhance dense representations could break down in such cases. Further study on these edge cases would be valuable.

---

> ### Author Rebuttal · Authors · 2025-07-30
>
> We sincerely appreciate the reviewer's thorough evaluation and constructive suggestions. Below we provide a comprehensive response addressing all concerns.
>
> ---
>
> ### **For Weakness 1: Baseline Comparison with LETTER**
>
> We appreciate the reviewer's suggestion to include Learnable Item Tokenization for Generative Recommendation (LETTER[1] ) as a baseline. We recognize LETTER's valuable contribution in proposing a learnable item tokenizer that incorporates regularization to capture hierarchical semantics, collaborative signals, and code assignment diversity within item identifiers. Our initial exclusion of LETTER as a direct baseline was primarily due to the distinct focus of our work. While LETTER optimizes the item tokenization process, enhancing the quality of discrete representations, COBRA introduces a cascaded sparse-dense paradigm. Our method's primary advancement lies in jointly modeling sparse ID generation and dense vector refinement, enabling end-to-end optimization and conditional generation where sparse IDs explicitly guide dense prediction.
>
> To comprehensively address this point, we conducted comparisons between COBRA and LETTER's implementations (LETTER-TIGER and LETTER-LC-Rec) on the Amazon Beauty dataset:
>
> | Method          | Recall@5 | Recall@10 | NDCG@5 | NDCG@10 |
> |-----------------|----------|-----------|--------|---------|
> | LETTER-TIGER    | 0.0431   | 0.0672    | 0.0286 | 0.0364  |
> | LETTER-LC-Rec   | 0.0505   | 0.0703    | 0.0355 | 0.0418  |
> | COBRA (Ours)    | 0.0537   | 0.0725    | 0.0395 | 0.0456  |
>
> Our approach achieves performance improvements of **7.9-38.1%** over LETTER-TIGER and **3.1-11.3%** over LETTER-LC-Rec across all metrics. This demonstrates the effectiveness of COBRA's cascaded sparse-dense generative approach.
>
> Furthermore, we consider LETTER's optimization of the item tokenization process to be an effective plug-and-play enhancement for semantic IDs, which does not conflict with our core contribution. In future work, COBRA could incorporate designs from LETTER to improve the generation process of sparse IDs, potentially yielding even better results.
>
> ---
>
> ### **For Weakness 2: Model Scalability**
>
> For public datasets, we primarily utilized a lightweight architecture (1-layer encoder, 2-layer decoder) to demonstrate efficiency. To address scalability concerns, we modified model parameters (2-layer encoder, 4-layer decoder) and conducted additional experiments on the Amazon Beauty dataset:
>
> | Model Scale         | Recall@5 | Recall@10 | NDCG@5 | NDCG@10 |
> |---------------------|----------|-----------|--------|---------|
> | COBRA-Base (1-enc, 2-dec)  | 0.0537   | 0.0725    | 0.0395 | 0.0456  |
> | COBRA-Large (2-enc, 4-dec) | 0.0535   | 0.0774    | 0.0384 | 0.0461  |
>
> Given the smaller scale of public datasets, further deepening the network does not yield significant benefits. This aligns with scaling law research[2-3] and our industrial experience, indicating that model size increases must correspond with data volume for gains. As supporting evidence, in larger-scale industrial scenarios, we have deployed a COBRA model with a more substantial architecture (6-layer encoder + 8-layer decoder), achieving significant benefits in industrial applications. These findings demonstrate that COBRA's cascaded paradigm can flexibly scale according to data volume, exhibiting architectural flexibility and real-world scalability.
>
> ---
>
> ### **For Weakness 3: Theoretical Analysis and Cold-Start Robustness**
>
> #### **Theoretical Justification for Cascaded Design**
>
> The reviewer raises an excellent point regarding the theoretical analysis of our cascaded design. Our approach addresses a key limitation of ID-based models like TIGER[4]  where discrete semantic IDs can limit expressiveness for fine-grained user intent. We augment each item with a dense representation to capture more comprehensive information, thus each item is represented by a (sparse, dense) pair. We theoretically demonstrate the superiority of our cascaded prediction modeling compared to an independent prediction approach.
>
> Let $S_{1:t}$ be a user's behavior sequence. For an item, $C$ represents a discrete random variable for its sparse ID, and $X$ represents a continuous random variable for its dense vector. We denote the information entropy of a discrete random variable as $H$ and the differential entropy of a continuous random variable as $h$. Our objective is to model $P(Item\_{t+1} | S\_{1:t})$ for predicting the next interacted item. We consider two distinct approaches for modeling $Item_{t+1}$:
>
> **Approach 1: Independent Prediction** Predict $C$ and $X$ separately based on the user's historical sequence $S\_{1:t}$. The probability distribution for this approach is formulated as:
>     $$P(Item_{t+1} | S_{1:t}) = P(C | S_{1:t}) \times P(X | S_{1:t}) \quad (1)$$
>
> **Approach 2: Cascaded Prediction** First predict the sparse information of the $Item_{t+1}$ using the user's historical sequence $S\_{1:t}$, and then utilize this sparse information along with the preceding sequence $S\_{1:t}$ to predict the dense vector of the next item. The probability distribution for this approach is formulated as:
>     $$P(Item_{t+1} | S_{1:t}) = P(C | S_{1:t}) \times P(X | C, S_{1:t}) = P(C, X | S_{1:t}) \quad (2)$$
>
> We demonstrate that the cascaded prediction modeling (Approach 2) consistently yields superior performance compared to independent prediction modeling.
>
> **Theorem 1: Superiority of Cascaded Modeling**
> Let $E_1$ be the information entropy of the independent prediction modeling probability distribution (Equation 1), and $E_2$ be the information entropy of the cascaded prediction modeling probability distribution (Equation 2). Then, it always holds that $E_2 \leq E_1$, where equality holds if and only if the random variables $C$ and $X$ are conditionally independent given $S\_{1:t}$.
>
> **Proof:**
> We calculate the information entropy for both probability distributions given by equations (1) and (2).
>
> For independent prediction modeling (Equation 1):
> $$
> \begin{aligned}
> E\_1 &= -\sum\_{c} \int_{x} P(c|S_{1:t}) \times P(x | S_{1:t}) \log(P(c|S_{1:t}) \times P(x | S_{1:t})) dx \\\\
> &= -\sum_{c} \int_{x} P(c|S_{1:t}) \times P(x | S_{1:t}) \log(P(c|S_{1:t})) dx \\\\
> &\quad -\sum_{c} \int_{x} P(c|S_{1:t}) \times P(x | S_{1:t}) \log(P(x | S_{1:t})) dx \\\\
> &= H(C | S_{1:t}) + h(X | S_{1:t})
> \end{aligned}
> $$
>
> Hence, under independent modeling, the total entropy decomposes additively over $C$ and $X$.
>
> For cascaded prediction modeling (Equation 2):
> $$
> \begin{aligned}
> E_2 &= -\sum_{c} \int_{x} P(c|S_{1:t}) \times P(x | c, S_{1:t}) \log(P(c|S_{1:t}) \times P(x | c, S_{1:t})) dx \\\\
> &= -\sum_{c} \int_{x} P(c, x|S_{1:t}) \log(P(c, x|S_{1:t})) dx \\\\
> &= H(C, X | S_{1:t})
> \end{aligned}
> $$
>
> Hence, cascaded modeling captures the full joint uncertainty of $C$ and $X$. By the subadditivity of information entropy[5]: $$E_2 = H(C, X | S_{1:t}) \leq H(C | S_{1:t}) + h(X| S_{1:t}) = E_1$$
> Equality holds if and only if random variables $C$ and $X$ are conditionally independent.
>
> Theorem 1 demonstrates that our cascaded modeling scheme always yields a probability distribution with lower information entropy. This implies that the resulting probability distribution is more concentrated, which is beneficial for model learning. Equality holds only when random variables $C$ and $X$ are conditionally independent, in which case $P(x|c, S_{1:t}) = P(x | S_{1:t})$. From equations (1) and (2), it is evident that both modeling approaches are equivalent under this condition. Consequently, cascaded modeling is generally superior to independent modeling.
>
> #### **Cold-Start Robustness Mechanisms**
>
> As noted, cold-start presents a challenging problem in recommendation systems. However, COBRA can effectively handle cold-start scenarios to a considerable extent.
>
>  **1) Content-Based Dense Representations for Cold-Start Items:**
>  COBRA processes raw item attributes (e.g., title, description) to generate dense vectors.
>  Crucially, this process operates independently of auto-incrementing system ID.
>  This ensures that even for cold-start items with no interaction history, meaningful dense representations are generated based purely on their content.
>
> **2) Leveraging Semantic IDs for Cold-Start Items:**
> For cold-start items, we generate sparse IDs using the same method as TIGER. As discussed in Section 4 of the TIGER[2] paper, compared to "Random IDs" used in traditional recommendation methods and KNN in semantic space, semantic IDs can effectively leverage content-based semantic information, ensuring that items still possess meaningful semantic ID even in cold-start scenarios.
>
> **3) Empirical Validation in Cold-Start Scenarios:**
> Moreover, according to our online A/B test results, cold-start advertisements achieved a **2.43% increase in conversion**, demonstrating the robustness of the COBRA method in cold-start scenarios.
>
> **4) Addressing Poorly Learned IDs:**
> The reviewer's concern regarding "poorly learned IDs" representing an extreme edge case is valid. While our end-to-end training framework fosters mutual learning between sparse and dense representations, inherently reducing the likelihood of such scenarios, the performance under poorly learned sparse IDs could be further analyzed. We plan to explore the effects of such extreme edge cases and potential mitigation strategies (e.g., adaptive weighting of sparse/dense contributions based on ID quality confidence) as part of our future work.
>
> ---
>
> ### References
> [1] Wang et al., "Learnable Item Tokenization for Generative Recommendation", CIKM 2024
> [2] Hoffmann et al., "Training Compute-Optimal Large Language Models", 2022
> [3] Hou et al., "Scaling Law of Large Sequential Recommendation Models", 2023
> [4] Rajput et al., "Recommender Systems with Generative Retrieval", NeurIPS 2023
> [5] Cover & Thomas. Elements of Information Theory. Wiley-Interscience, USA, 2006

---

> > ### Comment · Reviewer_XoPo · 2025-08-07
> >
> > The authors addressed some of my concerns in the rebuttal and demonstrated effectiveness in industrial scenarios, which led me to slightly raise my score. However, this paper is still largely engineering-focused with limited novelty. Moreover, in the rebuttal’s proof for “Weakness 3,” the authors claim that lower entropy facilitates learning, but in recommender systems, considering the need for diversity, lower entropy may not necessarily be beneficial. In addition, when addressing the scaling issue, the authors did not provide concrete improvements in real-world scenarios, which reduces the credibility of their claims.

---

> > > ### Author Response · Authors · 2025-08-08
> > >
> > > We sincerely thank the reviewer for the thorough reading of our rebuttal and for the insightful comments. The questions regarding the balance between information entropy and diversity, as well as the need for concrete evidence of industrial-scale gains.
> > >
> > > ### On Entropy, Learning, and Diversity
> > >
> > > From our perspective, the enhanced modeling accuracy resulting from low entropy does not conflict with achieving diverse recall results.
> > >
> > > Our cascaded design's low entropy signifies a more concentrated probability distribution for prediction, which is a key advantage for training. Our theoretical analysis demonstrates that this design more accurately captures the joint distribution over (sparse, dense) pairs. A tighter joint distribution yields lower entropy during training, contributing to more robust learning.
> > >
> > > We agree that diversity is a critical issue in recommender system. To ensure diversity, we introduce the BeamFusion mechanism in the inference stage. As detailed in  Figure 5(b) of our paper, BeamFusion uses tunable coefficients to control diversity.
> > >
> > > ### On Model Scalability and Industrial Applications
> > >
> > > In our industrial production environment, we evaluated the model under five different settings of user-interaction logs.
> > >
> > > | Model Size | Time Window | Recall@800 |
> > > |------------|-------------|------------|
> > > | 0.1 B      | 60 days     | 0.3870     |
> > > | 0.2 B      | 60 days     | 0.4466     |
> > > | 0.8 B      | 60 days     | 0.4531     |
> > > | 0.2 B      | 120 days    | 0.4544     |
> > > | 0.8 B      | 120 days    | 0.4928     |
> > >
> > > The results demonstrate that increasing the model size from 0.1B to 0.2B using 60 days of data led to 5.96% improvement in Recall@800.
> > > However, further scaling the model to 0.8B with the same 60-day window, or scaling the data window to 120 days with a 0.2B model, yielded diminishing gains.
> > > By contrast, simultaneously increasing both the model size and the data window (from 0.2B/60 days to 0.8B/120 days) resulted in a notable 4.62% improvement.

---

### Official Review · Reviewer_29r5 · 2025-07-03

**Clarity:** 3
**Significance:** 3
**Originality:** 2
**Rating:** 4
**Confidence:** 4

**Summary:**

This paper proposes COBRA, a generative recommendation framework that integrates sparse IDs for coarse-grained item semantics and dense vectors for fine-grained details, enabling precise and personalized recommendations. During inference, COBRA introduces a BeamFusion mechanism, combining beam search with nearest neighbor scores to enhance inference flexibility and recommendation diversity. Extensive experiments on both public and industrial datasets demonstrate the effectiveness, robustness, and scalability of COBRA.

**Questions:**

1. The paper claimed: "However, unlike [19], which uses a different configuration, we employ a 3-level semantic ID structure, where each level corresponds to a codebook size of 32" (Lines 227-228). What are the advantages of using the same codebook sizes across all levels, compared to varying codebook sizes?
2. COBRA uses 3-level semantic IDs (32×32×32) for public datasets (Lines 227–228), while it uses 2-level semantic IDs (32×32) for industrial-scale datasets (Line 254). Does this imply that the number of levels is dataset-dependent? If so, the paper should provide empirical evidence and a guideline for choosing appropriate configurations under different conditions. This also raises concerns about the COBRA’s adaptability across different settings.
3. In Section 4.2, COBRA uses 2-level semantic IDs (32×32), but its variant “COBRA w/o Dense” uses 3-level semantic IDs (256×256×256) (Line 260). What is the rationale for this difference? This setup may affect the fairness of the ablation.

**Ethical Concerns:**

["NO or VERY MINOR ethics concerns only"]

**Final Justification:**

The authors have addressed my comments, and I have maintained the original score (Borderline accept).

**Limitations:**

See the above **Weaknesses**.

**Quality:**

2

**Strengths And Weaknesses:**

**Strengths**

1. This paper integrates dense vectors to capture fine-grained details of items through end-to-end training, effectively mitigating the information loss in ID-based methods.
2. This paper presents a BeamFusion mechanism during inference, which balances recommendation accuracy and diversity.
3. Experimental results validate the COBRA's superiority over compared baselines, with detailed quantitative results and analyses.
4. This paper is well-organized and clearly written, with a clear introduction, related work, methodology, and experimental evaluation.

**Weaknesses**

1. While the paper incorporates dense vectors to enhance sequential recommendation accuracy over TIGER [1], it also introduces additional computational overhead. Therefore, providing a comparative analysis of model complexity and training/inference runtime across baselines would be valuable for assessing the effectiveness and efficiency of COBRA.
2. The paper lacks essential implementation details, including the embedding dimension, detailed architectures of the encoder and decoder, optimizer choices, learning rate, batch size, and pretraining settings for the residual quantization model (Figure 2). These omissions raise concerns about the reproducibility of the experimental results.
3. Several important baselines in the generative recommendation are missing, such as ColaRec [2], LC-Rec [3], and RPG [4]. Including comparisons with these methods would better highlight the effectiveness of the proposed COBRA.

4. While ablation studies are conducted on industrial datasets (Section 4.2), additional ablation analysis on public datasets would further validate the robustness and generalizability of the proposed COBRA.
5. Some typos are present.
6. In Lines 108 and 174, "encoder Encoder" should be revised for consistency in formatting—either uniformly bolded or unbolded; In Figure 1, the Transformer encoder should be revised—it should be either shown or omitted on both sides to avoid misleading readers into thinking the proposed COBRA omits the encoder compared to TIGER; In Line 170, $\bold{v}_{j}$ should be revised to $\bold{v}_{{item}_j}$; In Lines 153–155, obtaining $\bold{e}_{t+1}$ from predicted $ID_{t+1}$ via the embedding layer Embed (as defined in Line 136) should be explicitly described to ensure clarity in the methodology.

**Reference**

[1] Recommender Systems with Generative Retrieval, NeurIPS, 2023.

[2] Content-Based Collaborative Generation for Recommender Systems, CIKM, 2024.

[3] Adapting Large Language Models by Integrating Collaborative Semantics for Recommendation, ICDE, 2024.

[4] Generating Long Semantic IDs in Parallel for Recommendation, KDD, 2025.

---

> ### Author Rebuttal · Authors · 2025-07-30
>
> We thank the reviewer for their thorough feedback and constructive suggestions, which have significantly helped us improve the manuscript. We address each point below.
>
> ---
>
> ### **For Weakness 1: Computational Overhead**
>
> We appreciate this important concern. For our complexity analysis, we define $L$ as the sequence length, $T$ as the number of tokens per item, and $D$ as the embedding dimension.
>
> **Training Complexity**
> | Component | COBRA | TIGER[1] (GR) | SASRec[2] (DR) |
> | :---------------------- | :------------------------------------------- | :--------------------------- | :--------------------------- |
> | Item Feature Encoder | $\mathcal{O}(L \cdot T^2 \cdot D + L \cdot T \cdot D^2)$ | N/A | N/A |
> | Sequential Model | $\mathcal{O}(L^2 \cdot D + L \cdot D^2)$ | $\mathcal{O}(L^2 \cdot D + L \cdot D^2)$ | $\mathcal{O}(L^2 \cdot D + L \cdot D^2)$ |
>
> **Inference Complexity**
> | Component               | COBRA                     | TIGER                     | SASRec                    |
> |-------------------------|---------------------------|---------------------------|---------------------------|
> | Item Feature Encoder    | $\mathcal{O}(1)$ (cached) | N/A                       | N/A                       |
> | Sequential Model        | $\mathcal{O}(L^2 \cdot D + L \cdot D^2)$ | $\mathcal{O}(L^2 \cdot D + L \cdot D^2)$| $\mathcal{O}(L^2 \cdot D + L \cdot D^2)$|
>
> To ensure practical efficiency, we employ several key techniques during implementation. These include sequence Packing  and FlashAttention to minimize computational waste. Furthermore, caching is utilized to decouple the encoder computation, significantly speeding up the inference process. Thanks to these optimizations, COBRA achieves over 30% Model FLOPs Utilization, surpassing typical recommendation benchmarks[3]. This efficiency has enabled successful deployment, serving over 200 million daily users.
>
> ---
>
> ### **For Weakness 2: Missing Implementation Details**
>
> We apologize for the omission of critical implementation details. We will add a "Implementation Details" table in the Appendix. Below is a comprehensive list of all key hyperparameters:
>
> | Module | Public Datasets | Industrial Datasets |
> | :-------------------- | :------------------------------------------- | :------------------------------------------- |
> | Embedding dimension | 128 | 768 |
> | Encoder layers | 1 | 6 |
> | Decoder layers | 2 | 8 |
> | Attention heads | 8 | 12 |
> | Optimizer | AdamW | AdamW |
> | Learning rate | $3\mathrm{e}{-4}$  | $4\mathrm{e}{-5}$  |
> | Batch size | 64 | 64 |
> | Dropout | 0.1 | 0.1 |
> | RQ-VAE pretrain | T5-base, $32 \times 32 \times 32$ | ERNIE, $32 \times 32$ |
> | Training epoch | 10 | 1 |
>
> ---
>
> ### **For Weakness 3: Missing Baselines (ColaRec, LC-Rec, RPG)**
>
> We thank the reviewer for highlighting these recent and important generative recommendation baselines. We agree that a comprehensive comparison is valuable for positioning our work within the evolving landscape of the field.
>
> Regarding RPG[4] , we note its publication on arXiv in June 2025, which falls after our submission deadline. For ColaRec[5] and LC-Rec[6], their core contributions lie in aligning semantic IDs with collaborative information, often leveraging pre-trained graph-based representations. While conceptually related to generative recommendations, their distinct methodological focus on explicit semantic-collaborative alignment differs from COBRA's emphasis on unified cascaded sparse-dense representations through end-to-end training.
>
> However, to ensure a more comprehensive comparison, we have now conducted experiments on public datasets with LC-Rec. The results on the "Beauty" dataset are as follows:
>
> | Metric | Recall@5 | Recall@10 | NDCG@5 | NDCG@10 |
> | :------- | :------- | :-------- | :------- | :-------- |
> | LC-Rec | 0.0482 | 0.0681 | 0.0327 | 0.0409 |
> | COBRA | **0.0537** | **0.0725** | **0.0395** | **0.0456** |
>
> COBRA consistently outperforms LC-Rec across all metrics, with improvements ranging from 6.4% to 20.8%. This further validates the effectiveness of COBRA's cascaded sparse-dense representation approach for generative recommendation.
>
> ---
>
> ### **For Weakness 4: Ablation Studies on Public Datasets**
>
> We initially conducted ablation studies primarily on industrial datasets due to their direct relevance to our business objectives. To address the reviewer's concern about generalizability, we have now performed additional ablation analyses on the "Beauty," "Sports & Outdoors," and "Toys & Games" public datasets. The Recall@10 results are summarized below:
>
> | Method | Beauty | Sports & Outdoors | Toys & Games |
> | :---------------------- | :------- | :---------------- | :------------- |
> | COBRA w/o Dense | 0.0656 | 0.0331 | 0.0713 |
> | COBRA w/o ID | 0.0681 | 0.0365 | 0.0653 |
> | COBRA w/o BeamFusion | 0.0714 | 0.0413 | 0.0769 |
> | **COBRA (Full)** | **0.0725** | **0.0424** | **0.0781** |
>
> The results on public datasets align with those from the industrial setting, demonstrating that the absence of any key component (Dense, ID, or BeamFusion) leads to a performance drop. This confirms the robustness and generalizability of COBRA's proposed components across different scales and data characteristics.
>
> ---
>
> ### **For Weaknesses 5-6: Formatting and Typos**
>
> We appreciate you pointing out these errors. We have thoroughly reviewed the manuscript and made the following corrections:
> * The notation of $\text{Encoder}$ in line 108 have been changed to $\mathbf{Encoder}$ for consistency throughout the paper.
> * In Figure 1, both sides now explicitly show the  encoder to prevent any misunderstanding regarding COBRA's architecture.
> * In Line 170, the formula have been revised to $v_{{item}_j}$. We have also explicitly clarified in the text that "the sparse ID is mapped into a dense vector space through an embedding layer" in Section 3.1 to ensure clarity in the methodology.
>
> All these modifications will be incorporated into the final version of the paper.
>
> ---
>
> ### **For Question 1: Advantages of using the same codebook sizes across all levels**
>
> Using a uniform codebook size (32) across all three levels for public datasets offers design simplicity and streamlines quantization implementation. Based on current empirical observations, this configuration yields satisfactory results. However, a detailed investigation into the impact of varying codebook sizes remains an avenue for future research.
>
> ---
>
> ### **For Question 2: Number of levels being dataset-dependent and COBRA's adaptability**
>
> The optimal number of semantic ID levels is primarily driven by **Performance Requirements and Computational Cost**. In industrial scenarios, there's a critical demand for retrieving a large number of diverse items from vast candidate sets within strict latency constraints. Based on our practical experience, employing finer-grained ID hierarchies significantly increases the computational overhead of beam search, which can severely impact real-time performance. Conversely, for smaller datasets or scenarios with less stringent latency requirements, more hierarchical levels might be permissible to capture finer semantic distinctions if needed.
>
> COBRA demonstrates its adaptability by successfully operating with different semantic ID structures while achieving strong performance. This flexibility indicates that COBRA's core architecture effectively integrates these representations, allowing for configurations that balance semantic granularity with the practical demands of the deployment environment.
>
> ---
>
> ### **For Question 3: Rationale for COBRA w/o Dense using 3-level semantic IDs vs. COBRA's 2-level IDs on industrial dataset**
>
> COBRA's full model uses $32 \times 32$ IDs because dense representations provide significant fault tolerance, allowing for a coarser ID design.
>
> In contrast, "COBRA w/o Dense" **completely removes dense vectors**, necessitating **much finer-grained sparse IDs** to compensate for lost information. A $32 \times 32$ ID set  would be insufficient for the 2 million industrial advertisements, leading to substantial information loss and consequently poor recall. Therefore, a $256 \times 256 \times 256$ configuration is used for "COBRA w/o Dense" to maximize its representational capacity and enable a fair baseline comparison against the full model. This highlights the synergistic benefit of COBRA's cascaded sparse-dense approach, achieving superior performance with a less complex sparse ID structure.
>
> ---
>
> ### References
> [1] Rajput et al., "Recommender Systems with Generative Retrieval", NeurIPS 2023
> [2] Kang & McAuley, "Self-Attentive Sequential Recommendation", ICDM 2018
> [3] Zhou et al., "OneRec: Optimizing Recommendation Training Efficiency", Technical Report 2023
> [4] Hou et al., "Generating Long Semantic IDs in Parallel for Recommendation", KDD 2025
> [5] Wang et al., "Content-Based Collaborative Generation for Recommender Systems", CIKM 2024
> [6] Zheng et al., "Adapting Large Language Models by Integrating Collaborative Semantics for Recommendation", ICDE 2024

---

> > ### Comment · Reviewer_29r5 · 2025-08-07
> >
> > Thanks author's rebuttal. The authors have addressed my comments, and I will maintain the original score.

---

> > > ### Author Response · Authors · 2025-08-08
> > >
> > > ​​Thank you for your reply. We really appreciate your time and valuable insights once again.

---

### Official Review · Reviewer_XgPc · 2025-07-04

**Clarity:** 3
**Significance:** 2
**Originality:** 2
**Rating:** 4
**Confidence:** 3

**Summary:**

This paper proposes COBRA, a new model which integrates generative and dense retrieval method by formulating sparse ID generation and dense vector prediction through a cascading manner in sequential recommendation. COBRA is trained via optimizing both the CE loss of next token prediction for sparse IDs and the CE loss with in-batch negatives for dense vectors. During inference, COBRA leverages beam search to generate several distinct IDs and dense vectors, builds the candidate item set by combining items corresponding to these IDs and items retrieved by dense vectors through MIPS, and the final result is generated by selecting top-K items in the candidate set via a non-linear combination of beam search scores and cosine similarity scores. Experiments on public and private datasets demonstrate the effectiveness of COBRA.

**Questions:**

1. Experiment results do not verify the necessity and importance of the cascade bi-represented formulation in COBRA, which raises doubts about the novelty of the paper.  It seems the performance improvement comes from ensembling generative retrieval and dense retrieval methods. A natural question is, what is the performance of combining retrieved results of existing GR and DR methods through BeamFusion or other simple fusion strategy like linear combining beam search and NN scores?
2. As COBRA leverages multiple dense embeddings for NN search, current dense retrieval methods with multiple embeddings should also considered as baseline for fair comparison. Examples includes MIND[1] and ComiRec[2].
3. What is the ablation study result in the public dataset?
4. What is the value of $M$ corresponding to Table 1&2?

[1] Multi-Interest Network with Dynamic Routing for Recommendation at Tmall
[2] Controllable Multi-Interest Framework for Recommendation

**Ethical Concerns:**

["NO or VERY MINOR ethics concerns only"]

**Final Justification:**

The authors' response has addressed most of my concerns and I raise my rating accordingly.

**Limitations:**

yes

**Quality:**

2

**Strengths And Weaknesses:**

Strength:
1. Writing is good and the paper is easy to follow
2. Experimental results seems impressive

Weakness:
1. The evaluation of major contributions (cascade bi-represented formulation, coarse-to-fine generation) is not thorough.
2. Lack of analysis of time complexity of training and inference, which is important for retrieval models.

---

> ### Author Rebuttal · Authors · 2025-07-30
>
> We sincerely appreciate your constructive feedback and insightful comments on our paper. Thank you for acknowledging the clarity of our presentation and the strength of our experimental results. We have thoroughly addressed your concerns regarding contribution evaluation, baseline comparisons, time complexity, and implementation settings. Below is our response incorporating additional analyses and experiments.
>
> ---
>
> ### **For Weakness 1 & Question 1: Enhanced Evaluation of Core Contributions**
>
> To rigorously validate the necessity of our cascaded bi-represented formulation—which extends beyond a simple combination of generative retrieval (GR) and dense retrieval (DR)—we conducted comprehensive experiments on public datasets. Our experimental design compares two variants:
>
> | Variant                  | Architecture                                       |
> |--------------------------|---------------------------------------------------|
> | **Ensembling-Linear**    | Combining GR and DR using linear fusion strategy  |
> | **Ensembling-BeamFusion**| Combining GR and DR using BeamFusion              |
> | **COBRA (Full)**         | Complete cascaded architecture                    |
>
> Recall@10 performance across public datasets:
>
> | Method                | Beauty   | Sports & Outdoors | Toys & Games |
> |-----------------------|----------|-------------------|--------------|
> | Ensembling-Linear     | 0.0670   | 0.0379            | 0.0720       |
> | Ensembling-BeamFusion| 0.0664   | 0.0380            | 0.0722       |
> | **COBRA (Full)**     | **0.0725** | **0.0424**        | **0.0781**   |
>
> COBRA outperforms ensembling approaches in Recall@10 metrics. For example:
> - On *Beauty*: Ensembling-BeamFusion (0.0664) vs. COBRA (0.0725, +$9.2\\%$)
> - On *Sports & Outdoors*: Performance gap of $10.6\\%$ between Ensembling-Linear and COBRA
>
> This demonstrates that cascaded generation—where sparse IDs explicitly condition dense vector refinement—is fundamentally more effective than ensembling separate GR and DR methods. The joint training ensures dense vectors are semantically anchored to sparse IDs, avoiding ambiguous retrievals.
>
> ---
>
> ### **For Weakness 2: Time Complexity Analysis**
>
> We appreciate this important concern. For our complexity analysis, we define $L$ as the sequence length, $T$ as the number of tokens per item, and $D$ as the embedding dimension.
>
> **Training Complexity**
> | Component | COBRA | TIGER[1] (GR) | SASRec[2] (DR) |
> | :---------------------- | :------------------------------------------- | :--------------------------- | :--------------------------- |
> | Item Feature Encoder | $\mathcal{O}(L \cdot T^2 \cdot D + L \cdot T \cdot D^2)$ | N/A | N/A |
> | Sequential Model | $\mathcal{O}(L^2 \cdot D + L \cdot D^2)$ | $\mathcal{O}(L^2 \cdot D + L \cdot D^2)$ | $\mathcal{O}(L^2 \cdot D + L \cdot D^2)$ |
>
> **Inference Complexity**
> | Component               | COBRA                     | TIGER                     | SASRec                    |
> |-------------------------|---------------------------|---------------------------|---------------------------|
> | Item Feature Encoder    | $\mathcal{O}(1)$ (cached) | N/A                       | N/A                       |
> | Sequential Model        | $\mathcal{O}(L^2 \cdot D + L \cdot D^2)$ | $\mathcal{O}(L^2 \cdot D + L \cdot D^2)$| $\mathcal{O}(L^2 \cdot D + L \cdot D^2)$|
>
> To ensure practical efficiency, we employ several key techniques during implementation. These include sequence Packing  and FlashAttention to minimize computational waste. Furthermore, caching is utilized to decouple the encoder computation, significantly speeding up the inference process. Thanks to these optimizations, COBRA achieves over 30% Model FLOPs Utilization, surpassing typical recommendation benchmarks[3]. This efficiency has enabled successful deployment, serving over 200 million daily users.
>
> ---
>
> ### **For Question 2: Comparison with Multi-Interest Retrieval Methods**
>
> We appreciate your suggestion to include multi-interest retrieval baselines. To provide a more comprehensive evaluation beyond sequential paradigms (TIGER for GR, SASRec for DR), we conducted experiments comparing COBRA against state-of-the-art multi-interest methods. Following literature findings that ComiRec[2] generally outperforms MIND[1], we focus our comparison on ComiRec:
>
> | Method   | Recall@5 | Recall@10 | NDCG@5 | NDCG@10 |
> |----------|----------|-----------|--------|---------|
> | ComiRec  | 0.0205   | 0.0445    | 0.0105 | 0.0183  |
> | COBRA    | **0.0537** | **0.0725** | **0.0395** | **0.0456** |
>
> **Key Observations**
> 1. Consistent Superiority:
>    COBRA demonstrates significant performance advantages over ComiRec across all metrics, with particularly notable gains in early-position ranking (Recall@5 and NDCG@5).
>
> 2. Architectural Distinction:
>    While ComiRec generates multiple interest embeddings per user, COBRA's cascaded paradigm fundamentally differs by:
>    - Explicitly conditioning dense vector generation on sparse IDs
>    - Enabling mutual refinement of both representations through end-to-end training
>
> ---
>
> ### **For Question 3: Ablation Studies on Public Datasets**
>
> We initially conducted ablation studies primarily on industrial datasets due to their direct relevance to our business objectives. To address the reviewer's concern about generalizability, we have now performed additional ablation analyses on the "Beauty," "Sports & Outdoors," and "Toys & Games" public datasets. The Recall@10 results are summarized below:
>
> | Method | Beauty | Sports & Outdoors | Toys & Games |
> | :---------------------- | :------- | :---------------- | :------------- |
> | COBRA w/o Dense | 0.0656 | 0.0331 | 0.0713 |
> | COBRA w/o ID | 0.0681 | 0.0365 | 0.0653 |
> | COBRA w/o BeamFusion | 0.0714 | 0.0413 | 0.0769 |
> | **COBRA (Full)** | **0.0725** | **0.0424** | **0.0781** |
>
> The results on public datasets align with those from the industrial setting, demonstrating that the absence of any key component (Dense, ID, or BeamFusion) leads to a performance drop. This confirms the robustness and generalizability of COBRA's proposed components across different scales and data characteristics.
>
> ---
>
> ### **For Question 4: Experimental Settings Clarifications**
>
> **Beam Size ($M$) Configuration**:
> - **Public Datasets** (Table 1): $M=20$
>   Optimal for smaller item pools ($11K$-$18K$ items), balancing diversity and efficiency.
> - **Industrial Dataset** (Table 2): $M=50$
>   Necessary for large-scale scenarios to ensure broad interest coverage.
>
> ---
>
> ### Conclusion
>
> We have demonstrated COBRA's performance gains stem from its **cascaded sparse-dense generative paradigm**, not simply GR/DR ensembling. The consistent performance gap across datasets validates each component's necessity. Our comparison with multi-interest methods highlights the architectural uniqueness of our approach, while complexity analyses confirm its practical scalability.
>
> ---
>
> ### Revision Plan:
> We will incorporate all modifications below in the final version:
> 1. Complete ablation tables (public + industrial datasets)
> 2. Expanded complexity analysis section
> 3. Multi-embedding Retrieval baseline comparisons
> 4. Engineering optimization details
>
> Thank you for your valuable feedback, which has significantly strengthened our work.
>
> ---
>
> ### References
> [1] Li et al., "Multi-Interest Network with Dynamic Routing", CIKM 2019
> [2] Cen et al., "Controllable Multi-Interest Framework for Recommendation", KDD 2020
> [3] Rajput et al., "Recommender Systems with Generative Retrieval", NeurIPS 2023
> [4] Kang & McAuley, "Self-Attentive Sequential Recommendation", ICDM 2018
> [5] Zhou et al., "OneRec: Optimizing Recommendation Training Efficiency", Technical Report 2023

---

### Note · Authors · 2025-08-14

We sincerely thank the reviewers for their rigorous review and insightful feedback. We are grateful for their recognition of our paper's core contribution: a generative recommendation framework that integrates sparse IDs and dense vectors via a cascaded learning strategy.
Reviewers consistently commended the paper's quality, highlighting its clear writing, well-structured presentation, and impressive experimental results (Reviewers XgPc, 29r5, XoPo). They acknowledged that our core contribution—the cascaded integration of sparse IDs and dense vectors—effectively addresses the challenge of "mitigating the information loss in ID-based methods" (Reviewers 29r5, 7Wfj). The proposed BeamFusion mechanism was also noted for its ability to balance recommendation accuracy and diversity (Reviewers 29r5, XoPo). The effectiveness of our method, demonstrated by "strong performance gains" on public benchmarks, was further validated through rigorous offline experiments and real-world A/B tests (Reviewers 29r5, XoPo, 7Wfj).
In response to the reviewers' insightful questions, we have conducted supplementary experiments and analyses to provide deeper insights into the model's key strengths:
1. Extended Experimental Analysis: We have enriched the experiment section with comprehensive comparisons against additional state-of-the-art baselines, ablation studies on public datasets, and an analysis of the model's scaling performance.
2. Complexity and Implementation Details: We have added a detailed analysis of training and inference complexity, along with key implementation details, to demonstrate the model's efficiency and scalability in practical applications.
3. Theoretical Analysis and Robustness: We have provided a theoretical justification for the cascaded design and discussed the model's robustness in cold-start scenarios, directly addressing questions regarding methodology and generalization.

All new results consistently support our original findings, further strengthening the novelty and practical utility of this work.

---

### Decision · Program_Chairs · 2025-09-17

**Decision:**

Accept (poster)

**Comment:**

This work continues the recent important trend of generative sequential recommenders by combining both sparse and dense representations in a cascaded manner that the authors theoretically show to reduce entropy, which is claimed to yield a better model fit.  A reviewer questions whether lower entropy negatively impacts diversity; the authors provide further technical explanation in the discussion, which should be incorporated in the paper to clarify the technical claims.  The work empirically achieves SOTA results on public academic and industrial datasets.

Overall, while the methodology seems somewhat incremental and highly engineered, SOTA results are achieved, theoretical justification is provided and this is a timely contribution to a nascent field of generative recommendation models.  After the extensive rebuttal discussion, all reviewers (weakly) recommend NeurIPS acceptance and the AC agrees with the decision to accept.